# A Causal Framework for Aligning Metrics of Image Quality and Deep Neural Network Robustness

## Abstract

Image quality plays an important role in the performance of deep neural networks (DNNs) and DNNs have been widely shown to exhibit sensitivity to changes in imaging conditions. Large-scale datasets often contain images under a wide range of conditions prompting a need to quantify and understand their underlying quality distribution in order to better characterize DNN performance and robustness. Aligning the sensitivities of image quality metrics and DNNs ensures that estimates of quality can act as priors for image/dataset difficulty independent task models trained/evaluated on the data. Conventional image quality assessment (IQA) seeks to measure and align quality relative to human perceptual judgements, but here we seek a quality measure that is not only sensitive to imaging conditions but also well-aligned with DNN sensitivities. We first ask whether conventional IQA metrics are also informative of DNN performance. In order to answer this question, we reframe IQA from a causal perspective and examine conditions under which quality metrics are predictive of DNN performance. We show theoretically and empirically that current IQA metrics are weak predictors of DNN performance in the context of classification. We then use our causal framework to provide an alternative formulation and a new image quality metric that is more strongly correlated with DNN performance and can act as a prior on performance without training new task models. Our approach provides a means to directly estimate the quality distribution of large-scale image datasets towards characterizing the relationship between dataset composition and DNN performance.

## 1 Introduction

Ensuring the robustness of deep neural networks (DNNs) to real-world imaging conditions is crucial for safety- and cost-critical applications. Extensive research has shown that DNNs remain sensitive to natural distortions (Taori et al., 2020; Djolonga et al., 2021; Ibrahim et al., 2022; Geirhos et al., 2021) despite efforts to close the gap between performance on clean and naturally-distorted images. While much effort has focused primarily on the design and optimization of robust DNNs, there is now growing interest in developing a deeper understanding of how the properties of the image data itself influence robustness during training and evaluation (Ilyas et al., 2022; Lin et al., 2022; Pavlak et al., 2023).

As image datasets grow in size, the cost and feasibility of using human annotators to assess and annotate properties of each data point becomes intractable. With pre-training datasets for foundation models and other large-scale vision models approaching hundreds of millions to billions of images (Radford et al., 2021; Sun et al., 2017; Schuhmann et al., 2022), new automated methods are needed for quantitatively assessing dataset composition. In particular, since image quality (IQ) is known to impact DNN performance (Taori et al., 2020; Djolonga et al., 2021; Hendrycks & Dietterich, 2019; Ibrahim et al., 2022), this motivates the need for methods that can estimate the underlying quality distribution of large-scale image datasets prior to training or evaluating downstream DNNs. Here *quality* describes the absence of distortion but more generally relates to the ability to extract task-relevant information from the image. Image *quality* and *difficulty* are closely related where quality measures properties of the imaging conditions while difficulty involves content and composition in addition to the conditions. Effective measures of image quality should provide

insight into image/dataset difficulty independent of knowledge or assumptions about the particular downstream task models that will consume the data. For instance, when training data is skewed in favor of high-quality images, analysis of the quality distribution using IQ metrics may help justify and identify more aggressive data augmentation strategies during training to ensure task model robustness. Similarly, analyzing the quality distribution of evaluation datasets may find that the data is not sufficiently challenging/diverse which could lead to a false sense of task model robustness in downstream evaluations. *Our goal in this work is to provide a framework to identify and develop quality measures that can act as **priors** on DNN performance towards characterizing the relationship between dataset composition and DNN robustness.*

In this work, we focus specifically on *natural robustness* which considers how images are distorted due to real-world factors such as lighting, weather, sensor settings, and/or motion. Image quality assessment (IQA) metrics have been developed over several decades of research (Wang & Bovik; Xu et al., 2017; Wang, 2004; Agnolucci et al., 2024) and provide quantitative measures of quality calibrated with respect to human perceptual judgements. To the best of our knowledge, little work has been done to understand how these IQA metrics can help relate image difficulty and DNN performance. To make this connection explicit, we state our primary research question: **What is the extent of the relationship between IQ and DNN performance metrics?**

**Contributions**   To answer this question, our work makes the following contributions:

- Our primary contribution is a causal framework for analyzing the relationship between image quality and DNN performance in a range of IQA settings
- We use the framework to establish theoretically and empirically the independence of image quality and DNN performance under general conditions
- We identify specific conditions under which IQA metrics can be predictive of DNN performance
- We use the framework in the context of image classification tasks to develop a new task-guided IQA metric that enables quantitative assessments of image quality that are also predictive of downstream DNN task performance

## 2 RELATED WORK

**Image Quality Assessment**   Image quality assessment has been long-studied in the computer vision and image processing literature. Full Reference IQA (FR-IQA) (Wang, 2004; Zhang et al., 2011; 2018) assume the availability of a reference or "clean" image against which the test image is compared and the quality is measured. In contrast, the No Reference IQA (NR-IQA) (Mittal et al., 2012; Wang et al., 2022; Agnolucci et al., 2024) setting (aka Blind IQA) uses only features of the test image to estimate a quality score. In both settings, conventional IQA methods are calibrated and compared against human perceptual judgements of quality such as Mean Opinion Scores (MOS). These measures are task-agnostic and humans are not required to make judgements about the content of the image but only to measure subjective "quality" (typically on a scale of 1-5, Poor-High). This motivates our investigation into whether these metrics can also provide task-relevant assessments of image quality.

**Relationship of DNNs and human perception**   A key question of this work centers on whether IQA metrics calibrated against human MOS are sensitive to any of the same image features that DNNs use for downstream tasks. Outside of the IQA literature, prior works have shown differences in humans and DNNs in the context of shape/texture bias (Geirhos et al., 2018; Hermann et al., 2019), shortcut learning (Geirhos et al., 2020; Zech et al., 2018; Brown et al., 2023; Ong Ly et al., 2024), and error consistency (Geirhos et al., 2021; Wichmann & Geirhos, 2023), but the question remains open whether IQA metrics aligned with human MOS correlate with DNN performance.

**Dataset difficulty/pruning**   Our work is strongly motivated by the growing interest in automated methods for dataset analysis. In particular, new methods focus on dataset pruning (Tan et al., 2023; He et al., 2023; Abbas et al., 2024), identifying difficult/important examples (Kwok et al., 2024; Ilyas et al., 2022) or data slices (Eyuboglu et al., 2022; Chung et al., 2019; Sohoni et al., 2020; Chen et al., 2019), and dataset auditing for shortcuts (Pavlak et al., 2023). Other results have shown that understanding dataset composition matters for analyzing model robustness (Ibrahim et al., 2022; Drenkow & Unberath, 2023). Our work takes a positive step towards automated methods for analyzing the quality distribution of image datasets and establishing priors on DNN performance.

## 3 CAUSAL FRAMEWORK FOR IQA

We first provide a causal inference perspective on the IQA problem. We use causal directed acyclic graphs (DAGs) to illustrate our assumptions about the imaging generating process, quality metric, and performance metric as well as the interactions between all associated variables. This causal framework provides a means for identifying the specific conditions under which quality metrics are predictive of DNN performance.

**Preliminaries** We specify a causal DAG $\mathcal{G}$ via a set of nodes $\mathcal{V}$ and directed edges $\mathcal{E}$. To obtain the causal interpretation, directed edges imply a causal relationship such that for a variable/node $V \in \mathcal{V}$, $V$ is a function of its parents ($V = f_V(pa(V), U)$ where $U$ is an exogenous noise term).

For defining causal models in the IQA context, we start from a set of factors $A \in \mathcal{A}$ that capture the key variables in the data generating process affecting the image conditions (e.g., lighting, focal length, aperture, exposure, weather). Let $X \in \mathcal{X}$ be the resulting images, and for a task $T$, let $Y \in \mathcal{Y}$ be the label associated with $X$ for the task. For this work, we focus on classification tasks where $\mathcal{Y}$ consists of a discrete set of $K$ classes ($\mathcal{Y} = \{1, \dots, K\}$). Our quality metric $Q : \mathcal{X} \to \mathbb{R}$ maps images to real number scores (typically in $[0, 1]$ where 1 is the highest quality). We also assume a downstream task DNN $f_\theta : \mathcal{X} \to \mathcal{Y}$ that maps images to class probabilities and is parameterized by $\theta$. We write the predicted probabilities $\hat{Y} = f_\theta(X)$ where $\hat{Y} \in \mathbb{R}^K$. Given one-hot encoded labels $Y$ and predictions $\hat{Y}$, we can compute a performance metric (e.g., accuracy) $M : \mathcal{Y} \times \mathcal{Y} \to \mathbb{R}$. In the general case and without loss of generality, we assume that $\hat{Y}$ is the prediction from a deterministic DNN. Similarly, we also assume that $Q, M$ are both deterministic functions of their parents in the causal DAG.

### 3.1 IQ METRIC DESIDERATA

Our primary motivation is to identify image quality metrics that allow us to assess the distribution of image quality in large-scale datasets and establish quality-driven priors for DNN performance independent of any specific trained task models. We propose the following desiderata for IQ metrics towards achieving these objectives.

- **D1 - Sensitive**: IQ metrics should be sufficiently sensitive to changes in image conditions
- **D2 - Blind**: IQ metrics should work in No Reference IQA (NR-IQA) settings where images are assessed *without* knowledge of a reference image captured under "clean" settings
- **D3 - Predictive**: IQ metrics should be correlated with DNN task performance
- **D4 - Task Model Agnostic**: IQ metrics should be designed/trained/calibrated without *a priori* knowledge of the downstream DNN models/architectures to be trained or evaluated on the data under consideration

The first criterion (**D1**) is a baseline condition requiring that the metric is actually sensitive to the natural conditions likely in the imaging domain. **D2** operates under the assumption that real-world datasets will not consist of pairs of clean/distorted images and will instead contain images collected in diverse conditions. **D3** stems from the idea that $Q$ should measure general properties of the data that influence $M$ (e.g., if $Q$ decreases, then $M$ should also decrease, although not necessarily at the same rate). Lastly, **D4** comes from the desire to use $Q$ to assess the composition of the dataset *independent* of any task-specific model training and without making assumptions about the type of DNN to be trained downstream. In other words, we want to avoid IQ metrics that are biased towards specific task models and/or require pre-training on each dataset to be analyzed.

### 3.2 BASELINE IQA FORMULATION

We start with the baseline formulation of the IQA problem as shown in Figure 1. We assume the labels $Y$ are determined from interpreting $X$ and that an oracle labeling function exists such that $Y$ can always be determined from $X$.

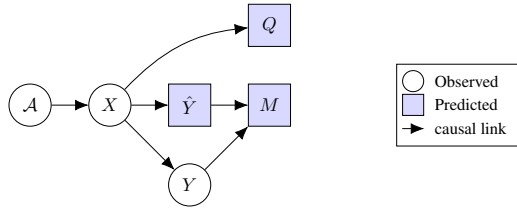

This model is a general formulation and makes no assumptions about the nature of the functions

Figure 1: Causal diagram relating model accuracy ($M$) with IQ metrics ($Q$).

that compute $Q, \hat{Y}$. This causal model is also consistent with conventional NR-IQA settings (Mittal et al., 2012; Ma et al., 2018) where the determination of $Q$ given $X$ is based on a function calibrated to human perceptual judgement and without knowledge of the task or labels.

**Conditional independence of** $Q, M$**:** The causal graph of Figure 1 illustrates that under the baseline formulation, $Q \perp M | X$ and $X$ is said to *d-separate* $Q, M$. The interpretation here is that given any image $X$, there is no expected relationship between $Q$ and $M$ by construction. This is not to say that a relationship cannot exist, but simply that there is nothing in this model that ensures it directly.

In addition to observing the d-separation of $Q, M$ by $X$, we can also compute the average causal effect (ACE) of $Q \to M$. We use the potential outcomes notation $M(Q = q)$ (or $M(q)$) to indicate the value of $M$ if $Q$ had been set to the value of $q$.

$$
\begin{aligned}
ACE(Q \to M) = \mathbb{E}[M(Q = q) - M(Q = q')] = \mathbb{E}[M(q)] - \mathbb{E}[M(q')] & \qquad \text{linearity of expectation} \\
= \mathbb{E}_X[\mathbb{E}[M(q)|X] - \mathbb{E}[M(q')|X]] & \qquad \text{law of total expectation} \\
= \mathbb{E}_X[\mathbb{E}[M(q)|Q = q, X] - \mathbb{E}[M(q')|Q = q', X]] & \qquad \text{unconfoundedness} \\
= \mathbb{E}_X[\mathbb{E}[M|Q = q, X] - \mathbb{E}[M|Q = q', X]] & \qquad \text{consistency} \\
= \mathbb{E}_X[\mathbb{E}[M|X] - \mathbb{E}[M|X]] = 0. & \qquad \text{conditional independence}
\end{aligned}
$$

The absence of causal effect and association between $Q, M$ in this formulation suggests that, without further assumptions, traditional IQA metrics should not be predictive of DNN performance. Furthermore, while No Reference and Full Reference (FR) IQA differ in their assumptions and setup, we provide their causal models as special cases of Figure 1 in Appendix A to show that $Q \perp M | X$ holds in both cases.

### 3.3 SHARED FEATURES IQA FORMULATION

Ideally, we would like $Q$ and $M$ to become dependent when conditioning on $X$ (i.e., we can learn about $M$ by observing only $Q$). This occurs in the case where there exists a common set of features $Z$ that are utilized both for the prediction function of $\hat{Y}$ and the quality score $Q$ as shown in Figure 2. This scenario does not presume a singular set of $Z$ that serves the task model and quality metric, but rather, the $Z$ shown here represents the intersection of features used by both. The existence of $Z$ ensures $Q, M$ are no longer independent given $X$. The primary question is whether such a $Z$ exists or whether $Q, M$ are related only as shown in Figure 1. An expanded discussion relating the baseline and shared features models can be found in Appendix B.

**Remark: Correlation vs. Causation** While we typically use causal models to estimate cause-effect relationships, a key clarification here is that we seek a weaker criterion, namely to establish the conditions under which quality and performance are *at least* correlated. Since we know that $X$ is a common cause for both $Q, M$, we first want to ensure that when we estimate $Q, M$ from $X$ we know the conditions under which $Q, M$ will be related via the same features of $X$ (e.g., via $Z$).

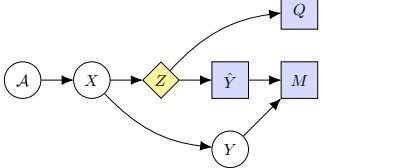

Figure 2: Causal diagram relating model accuracy ($M$) with IQ metrics ($Q$). In this case, $Q$ and $M$ are related via a common cause $Z$ that represents features in the image $X$ that influence both the prediction and quality.

## 4 WHAT IS THE RELATIONSHIP BETWEEN NR-IQA AND DNN PERFORMANCE METRICS?

Given the causal interpretation of IQA in §3, we now examine how conventional NR-IQA metrics relate to DNN performance. Our primary hypothesis is that if $Q, M$ are sensitive to a common set of visual features $Z$ derived from $X$ (Fig. 2), then we should observe that $Q$ is correlated with $M$ and even predictive of $M$ given $X$. Plainly stated, if image quality is high in general, then DNN performance should be similarly high (and vice versa).

We focus the following experiments on image classification tasks since they have available benchmark datasets and have been well-studied within the deep learning field. We show how our framework can

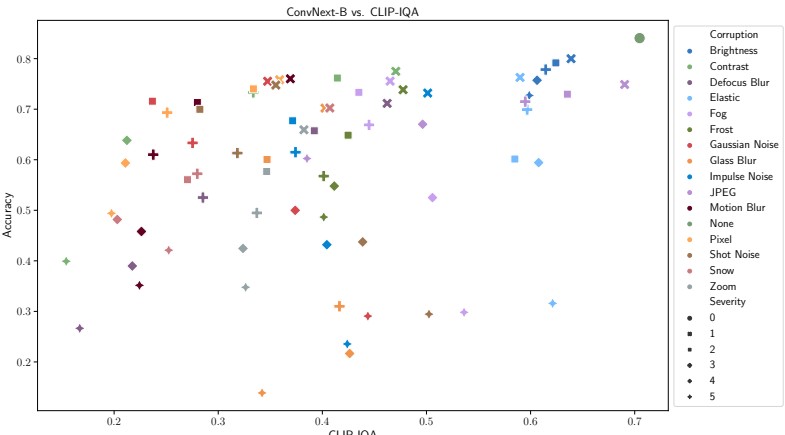

Figure 3: Accuracy ($M$) vs. IQ ($Q$) for ConvNext-B and CLIP-IQA respectively. Each point represents the average accuracy over all images in the ImageNet val set corrupted with the corresponding corruption/severity. Little correlation is observed between $M, Q$ across all corruptions/severities and $Q$ is weakly predictive of $M$.

be used to identify the relationship between IQA metrics and DNN performance as well as how it can guide the development of new metrics that satisfy all desiderata. For image classification, our experiments show that common NR-IQA methods are very weakly predictive of DNN performance, and while they satisfy **D1, D2, D4** of our IQ desiderata, they fail to satisfy **D3** and may not be suitable for estimating priors on DNN performance.

### 4.1 EXPERIMENT SETUP

In the experiments in this and subsequent sections, we use the following basic setup. In order to have precise control and knowledge of the type and severity of image distortion, we use the ImageNet validation (IN-val) and ImageNet-C (IN-C) (Hendrycks & Dietterich, 2019) datasets for evaluating IQ/performance on clean and corrupted images respectively. For reference, we provide a common corruptions causal DAG in Appendix C for comparison with the one in Figures 1 and 2.

For each experiment, we compute the IQ score ($Q$) and DNN correctness ($M$) for each image of the IN-C evaluation dataset. We use the following common and high-performing NR-IQA metrics ($Q$): CLIP-IQA (Wang et al., 2022), ARNIQA (Agnolucci et al., 2024), BRISQUE (Mittal et al., 2012), and Total Variation (TV). Here, CLIP-IQA and ARNIQA represent the state-of-the-art in learning-based IQA metrics while BRISQUE and TV represent conventional non-deep learning baselines. For DNNs, we evaluate the correctness ($M$) using pretrained ResNet34 (He et al., 2016), ConvNext-B (Liu et al., 2022), Swin-B (Liu et al., 2021) models provided via the `torchvision` package (Marcel & Rodriguez, 2010). Across all experiments, 95% confidence intervals (CI) are obtained via bootstrapping with 1000 resamples.

### 4.2 CORRELATION AND PREDICTABILITY OF $Q, M$ (**D3**)

We start by examining the correlation between $Q, M$ for NR-IQA metrics. Figure 3 shows the general relationship between $Q, M$ where each point in the figure is the average accuracy (over 50k images) for each corruption and severity in IN-C. Similarly, Table 1 computes the Kendall Rank Correlation Coefficient (KRCC), Spearman Rank Correlation Coefficient (SRCC), and Pearson Linear Correlation Coefficient (PLCC) between IQ and average accuracy across all corruption/severity pairs (75 total).

These results provide a look at group-wise association between $Q, M$ and the groups capture general trends in performance/IQ based on corruption type and severity. The low correlation between $Q, M$ suggests that these NR-IQA metrics likely fall under the model described by Figure 1 where $Q, M$ are conditionally independent given $X$.

Table 1: Correlation between IQ and accuracy, correctness. SRCC, PLCC computed using average accuracy for each (corruption, severity). AUC and CE based on point-wise predictions (95% CI within $\pm 0.001$). SRCC, PLCC values have $p < 0.05$.

| Model | IQA Metric | AUC ↑ | CE ↓ | $\|KRCC\|$ ↑ | $\|PLCC\|$ ↑ | $\|SRCC\|$ ↑ |
|---|---|---|---|---|---|---|
| ConvNext-B | ARNIQA | 0.517 | 0.677 | 0.088±0.149 | 0.168±0.215 | 0.127±0.214 |
| | BRISQUE | 0.568 | 0.670 | 0.255±0.129 | 0.398±0.190 | 0.374±0.182 |
| | CLIP-IQA | 0.567 | 0.670 | 0.273±0.154 | 0.328±0.202 | 0.378±0.212 |
| | TV | 0.477 | 0.676 | 0.108±0.183 | 0.138±0.294 | 0.151±0.255 |
| ResNet34 | ARNIQA | 0.499 | 0.663 | 0.003±0.155 | 0.006±0.205 | 0.007±0.225 |
| | BRISQUE | 0.552 | 0.658 | 0.175±0.140 | 0.278±0.186 | 0.254±0.202 |
| | CLIP-IQA | 0.599 | 0.647 | 0.307±0.159 | 0.467±0.188 | 0.429±0.207 |
| | TV | 0.500 | 0.657 | 0.051±0.194 | 0.291±0.256 | 0.047±0.264 |
| Swin-B | ARNIQA | 0.510 | 0.675 | 0.069±0.155 | 0.118±0.230 | 0.098±0.222 |
| | BRISQUE | 0.574 | 0.667 | 0.291±0.131 | 0.443±0.183 | 0.426±0.178 |
| | CLIP-IQA | 0.571 | 0.667 | 0.290±0.161 | 0.361±0.199 | 0.410±0.211 |
| | TV | 0.485 | 0.674 | 0.090±0.180 | 0.153±0.299 | 0.123±0.255 |

We also examine the point-wise relationship between $Q, M$. We aggregate DNN predictions and IQ values for all images in IN-C across all corruptions/severities and then randomly split the dataset (by image ID) into 80% training and 20% testing. We train a logistic regression classifier to predict $P(M|Q)$ and test on the hold-out set. We measure the predictability of $M$ using Area Under the Curve (AUC) and average cross-entropy (CE).

Table 1 shows that at the per-image level, $Q$ is still weakly predictive of $M$ (i.e., AUC $\approx 0.5$). This result is consistent with the theoretical analysis in §3 and the weak correlation observed empirically between $Q, M$ measured at the group level. While the causal DAG in Figure 1 would suggest that conditioning on $Y$ should not change the result, we test this empirically as follows.

We re-run the logistic regression for each label value in $\mathcal{Y}$ separately (1000 total) and compute the mean AUC ($mAUC$) and CE ($mCE$) across all labels. While we observe some variability in results when fixing $Y$, we find $mAUC = 0.5652$ ($\sigma = 0.08$) and $mCE = 0.6176$ ($\sigma = 0.1094$) suggesting that even when we control for $Y$, the predictability of DNN performance from the NR-IQA metrics remains weak.

These results suggest that NR-IQA metrics are likely sensitive to a different set of image features than task DNNs (i.e., no shared $Z$) and thus are barely, if at all, predictive of performance (i.e., they do not satisfy criterion **D3** from §3). The primary implication of this result is that if we intend to use IQ metrics to measure image quality/difficulty from the DNN perspective, common NR-IQ metrics may not be well-suited to this task and alternative approaches are needed.

## 5 RESTORING THE ASSOCIATION BETWEEN $Q, M$ VIA STRONG TASK-GUIDANCE (**D3**)

The previous results indicated that existing NR-IQA meet **D1, D2, D4** but the lack of predictability (**D3**) between NR-IQA metrics and DNN accuracy/correctness is a major limitation in using these metrics for assessing dataset quality relative to potential downstream task models. Focusing specifically on **D3**, we next consider an alternative formulation of the causal model will allow us to recover a dependence between $M, Q$ when conditioning on $X$.

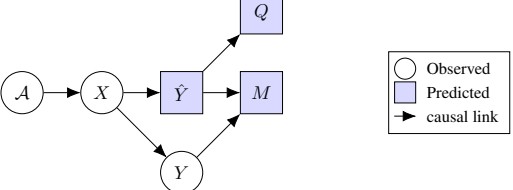

Figure 4: Causal diagram relating model accuracy ($M$) with IQ metrics ($Q$) with the additional dependence $\hat{Y} \rightarrow Q$.

In the case where a pre-trained DNN $f_\theta$ is given, Figure 4 describes a scenario where the predictions from this DNN may also be used as indicators of quality. This parallels other work (Hendrycks et al., 2019) which shows that uncertainty in the output predictions is often a good predictor of the OOD nature of the input. Note here that while $Q, M$ both depend on $\hat{Y}$, $Q$ requires no knowledge of the labels. In this case, it is possible that $\hat{Y}$ can be incorrect from the perspective of the ground truth label $Y$ but still provide information about $Q$ (e.g., via a low confidence prediction).

Table 2: Correlation between IQ and accuracy, correctness. KRCC, SRCC, PLCC computed using average accuracy for each (corruption, severity). AUC and CE based on point-wise predictions (95% CI within $\pm 0.001$). KRCC, SRCC, PLCC values have $p < 0.05$. Full table in Appendix E.

| Model | IQA Metric | AUC $\uparrow$ | CE $\downarrow$ | $\|KRCC\| \uparrow$ | $\|PLCC\| \uparrow$ | $\|SRCC\| \uparrow$ |
|---|---|---|---|---|---|---|
| | $Q_h$ | 0.772 | 0.562 | 0.660±0.070 | 0.822±0.070 | 0.854±0.063 |
| ConvNext-B | $Q_l$ | 0.778 | 0.555 | 0.660±0.067 | 0.826±0.067 | 0.854±0.063 |
| | $Q_p$ | 0.826 | 0.504 | 0.738±0.045 | 0.888±0.045 | 0.910±0.044 |
| | $Q_h$ | 0.848 | 0.470 | 0.862±0.028 | 0.930±0.028 | 0.969±0.023 |
| ResNet34 | $Q_l$ | 0.827 | 0.492 | 0.870±0.015 | 0.951±0.015 | 0.973±0.020 |
| | $Q_p$ | 0.850 | 0.461 | 0.886±0.015 | 0.960±0.015 | 0.977±0.021 |
| | $Q_h$ | 0.766 | 0.578 | 0.532±0.207 | 0.483±0.207 | 0.654±0.174 |
| Swin-B | $Q_l$ | 0.732 | 0.597 | 0.485±0.203 | 0.458±0.203 | 0.611±0.181 |
| | $Q_p$ | 0.807 | 0.529 | 0.603±0.184 | 0.620±0.184 | 0.732±0.142 |

Because this approach uses a model for $Q$ that is already trained for the classification task, we consider this **strong** task-guided IQA (TG-IQA). Clearly, this provides an alternative to the conventional NR-IQA metrics but now violates **D4** since $Q$ is informed directly by the same model trained for the task and measured by $M$. Nonetheless, our (temporary) goal here is to use the causal framework to show there exists a case where $Q, M$ are associated through a common set of features $Z$. Our hypothesis is that with **strong** TG-IQA we should observe a clear correlation between $M, Q$.

We examine the case where $Q$ is determined directly from predictions generated by a pre-trained task DNN. In this case, let $f_\theta$ be pre-trained to predict $P(Y|X)$. Then, let $z \in \mathbb{R}^k$ be the pre-softmax logits obtained from $f_\theta$ and $\hat{y} = \text{softmax}(z)$ where each $\hat{y}_i = P(Y = i|X)$ for $i \in 1, \ldots, K$. We consider three possible variants of $Q$ in this setting: (1) Max probability: $Q_p := \max_i \hat{y}_i$, (2) Entropy: $Q_h := H(\hat{y}) = -\sum_i \hat{y}_i \log \hat{y}_i$, and (3) Max logit: $Q_l := \max_i z_i$. While all three cases are inherently tied to the underlying label set $\mathcal{Y}$, the values of $Q$ do **not** have access to the ground truth label $Y$. Each of these $Q$ implicitly capture a DNN's confidence about its prediction and the natural underlying hypothesis is that confidence and image quality are positively correlated (i.e., as quality decreases, confidence also tends to decrease). These choices for $Q$ are driven by their use in out-of-distribution (Hendrycks & Gimpel, 2016; Hendrycks et al., 2019; 2020) and distribution shift detection (Wang et al., 2020).

### 5.1 EXPERIMENT - STRONG TASK-GUIDED IQA

Using the same setup as in Section 4.1, we now replace the NR-IQA metrics with $Q_p, Q_h, Q_l$. As in Section 4.2, we examine the group-wise correlation and point-wise predictability of $M$ from $Q$. To ensure our test of predictability is fair, we use separate models for obtaining $M$ and $Q$ (namely, ConvNext-B and Swin-B respectively). We provide additional results for other model pairs in Appendix E. Figure 5 shows the group-wise relationship between $Q, M$ where groups are averages over all images for the corresponding corruption, severity.

The results in Figure 5 and Table 2 show that strong task-guidance for $Q$ results in high correlation between $Q, M$ and predictability of $M$ from $Q$ (**D3**). This result, while expected, is important to show that using the causal framework it is possible to find a metric $Q$ that relies on a similar set of features as separate task models. However, like the previous section, this approach is only a partial solution as it satisfies **D1, D2, D3** but clearly violates **D4** by requiring a model already trained for the classification task.

## 6 RESTORING THE ASSOCIATION BETWEEN $Q, M$ VIA WEAK TASK-GUIDANCE (D3, D4)

So far, §4 showed that common NR-IQA metrics are weakly predictive of DNN performance and are therefore not viable candidates for supporting image/dataset-level analysis given our desiderata (§3). Then, we were able to address the predictability issue (**D3**) in §5 using strong task-guidance, but at the cost of requiring a task model already trained for the classification task (a violation of **D4**).

To address the aforementioned issues, we consider instead a weaker form of task-guidance where quality metrics can be aligned with task-specific information without requiring the expense of training

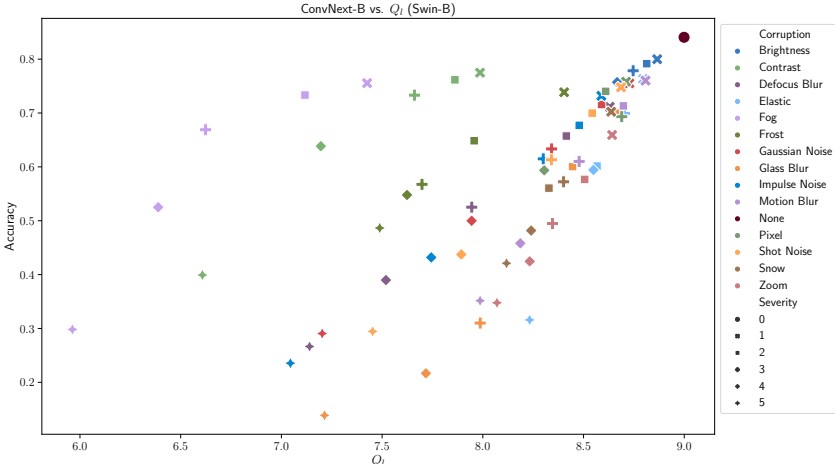

Figure 5: Accuracy vs. max logit using ConvNext-B for the task model and Swin-B for $Q_l$ which generally outperforms the other variants

a new task model directly on the dataset of interest (**D4**). In this setting shown in Figure 6, the computation of $Q$ is dependent not only on the image $X$ but on the label set $\mathcal{Y}$. The task $T$ is used as a selection variable (Zadrozny, 2004; Bareinboim et al., 2022) on which the dataset is conditioned, and as a collider in the DAG, $T$ creates an association between $M, Q$.

## 6.1 ZERO-SHOT CLIP IQA

We propose a quality metric that uses Zero-Shot (ZS) capabilities of the multi-modal CLIP foundation model (Radford et al., 2021) in order to address all desiderata (§3). In particular, we derive a new image quality metric (ZSCLIP-IQA) based on a zero-shot classification problem for our data and task of interest.

Let $\mathcal{D} = \{(x_i, y_i)\}_{i=1}^N$ be our dataset with images $x$ and labels $y$, $f : \mathcal{X} \rightarrow \mathcal{Z}$ be our CLIP image embedding network ($\mathcal{Z} \in \mathbb{R}^d$), and $g : \mathcal{T} \rightarrow \mathcal{W}$ be our CLIP text/token-embedding network ($\mathcal{W} \in \mathbb{R}^d$).

We define a set of task-relevant classes/tokens $\mathcal{T} = \{T_i\}_{i=1}^K$ that capture the text labels for concepts or entities likely to occur in the images (e.g., $K = 1000$ classes in the ImageNet dataset). We embed each of the text tokens $g(T_i) = w_i$ and normalize to get a unit vector representation for each token. Note when using CLIP as our text embedding network, we may also augment $T_i$ to include additional words (e.g., "A picture of a <token>"). The full set $\mathbf{W} = [w_0; w_1; \cdots w_K] \in \mathbb{R}^{d \times K}$ constitutes the ZS weights.

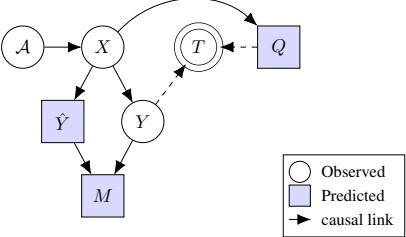

Figure 6: Causal DAG relating model performance ($M$) with IQ metrics ($Q$). $Q$ uses information about the label set $\mathcal{Y}$. The task $T$ is viewed as a selection variable which influences both the labels $Y$ and $Q$.

To evaluate image quality, we compute image embeddings $f_\theta(x) = z$ for $x \in \mathcal{D}$ which we normalize to be unit length. For each image, we compute the cosine similarity between the image embedding and each of the tokens $s = z\mathbf{W}$ (with $z \in \mathbb{R}^d$, $s \in \mathbb{R}^K$) and compute estimated class probabilities via a softmax over similarity scores ($\hat{y} = \text{softmax}(s)$). As in the strong TG-IQA scenario, we implement three variants of $Q$: (1) Max probability: $Q_p := \max_i \hat{y}_i$, (2) Entropy: $Q_h := H(\hat{y}) = -\sum_i \hat{y}_i \log \hat{y}_i$, and (3) Max-logit: $Q_l := \max_i s_i$.

## 6.2 EXPERIMENT - WEAK TASK-GUIDED IQA

Using the setup from §4 we now replace the NR-IQA metrics with $Q_p, Q_h, Q_l$ based on the ZSCLIP-IQA method described above. We again examine the group-wise correlation and point-wise pre-

Table 3: Correlation between IQ and accuracy. KRCC, SRCC, PLCC based on average accuracy for each (corruption, severity) combination. AUC and CE are computed based on point-wise predictions and all have 95% CI within $\pm 0.001$. All KRCC, SRCC, PLCC values have $p < 0.005$. Highlighted cells are maximum values over all models and IQA variants per column.

| Model | ZSCLIP-IQA | AUC ↑ | CE ↓ | $\mid KRCC \mid$ ↑ | $\mid PLCC \mid$ ↑ | $\mid SRCC \mid$ ↑ |
|---|---|---|---|---|---|---|
| | $Q_h$ | 0.349 | 0.677 | 0.738±0.067 | 0.869±0.059 | 0.906±0.048 |
| ConvNext-B | $Q_l$ | 0.675 | 0.632 | 0.573±0.084 | 0.764±0.080 | 0.783±0.085 |
| | $Q_p$ | 0.602 | 0.677 | 0.788±0.056 | 0.884±0.048 | 0.937±0.034 |
| | $Q_h$ | 0.666 | 0.663 | 0.800±0.059 | 0.936±0.028 | 0.945±0.032 |
| ResNet34 | $Q_l$ | 0.692 | 0.609 | 0.630±0.087 | 0.814±0.055 | 0.820±0.084 |
| | $Q_p$ | 0.368 | 0.663 | 0.834±0.049 | 0.949±0.024 | 0.959±0.026 |
| | $Q_h$ | 0.354 | 0.676 | 0.735±0.072 | 0.864±0.063 | 0.904±0.054 |
| Swin-B | $Q_l$ | 0.672 | 0.633 | 0.538±0.094 | 0.718±0.096 | 0.743±0.098 |
| | $Q_p$ | 0.601 | 0.676 | 0.778±0.059 | 0.879±0.050 | 0.935±0.034 |

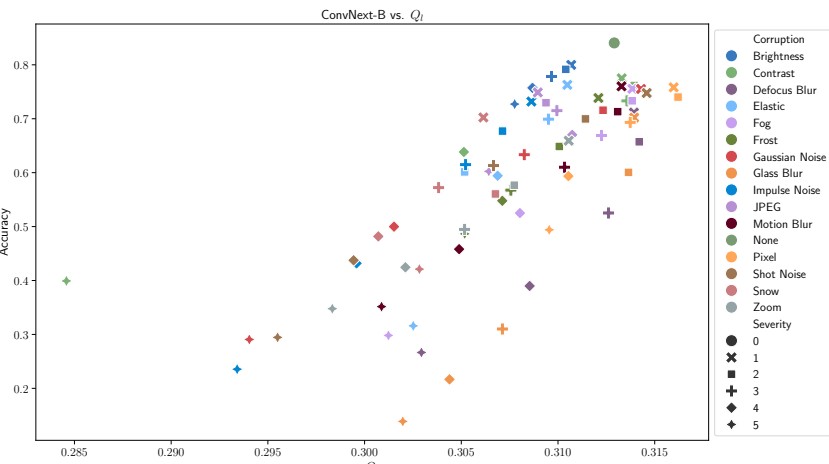

Figure 7: Accuracy vs. IQ with ConvNext-B as the task model $M$ and ZSCLIP-IQA max-logit as the quality metric $Q_l$ which generally outperforms the other variants.

dictability of $M$ from $Q$. Figure 7 and Table 3 show that weak task-guidance is enough to restore the association between $Q$ and $M$ without requiring a new task model to be trained on the dataset of interest.

**Remark:** While the CLIP backbone is pre-trained on a self-supervised task that resembles classification, it was not exposed to ImageNet (or IN-C) data during its training (see §5 in (Radford et al., 2021)) and can be effectively used here in a zero-shot setting to satisfy **D4**. In fact, while methods like CLIP-IQA and ARNIQA also rely on pre-trained backbones, the results of Tables 1 and 3 show that only ZSCLIP-IQA is "guided" (via our causal framework) to be a stronger predictor of DNN performance compared to other methods calibrated to human perceptual judgement.

## 6.3 EXPERIMENT - PREDICTABILITY OF DNN PERFORMANCE FOR MILDLY CORRUPTED DATASETS

In the previous experiments, the use of IN-C allowed us to investigate the large-scale effect of image corruptions on the predictability of performance by using multiple corrupted versions of the validation set with multiple levels of severity. In real-world datasets, we expect that only a small fraction of images will be corrupted. We next examine the extent to which IQA metrics can be used to show differences between the quality distributions of datasets containing varying levels of corruption while still satisfying **D1-D4** in these more realistic settings.

To answer this question, we generate new variants of IN-val consisting of mixtures of clean and corrupted images. For each variant, we specify a set of valid corruptions $\mathcal{C}$, severities $\mathcal{S}$, and a corruption probability $p_c$. We choose a fraction $1 - p_c$ of the original IN-val image IDs to remain as

clean images and a fraction $p_c$ to be corrupted. The corrupted images are sampled uniformly amongst the corruptions $c \in \mathcal{C}$ and severities $s \in \mathcal{S}$. The resulting variant consists of the original 50k image IDs with a mixture of clean and corrupted images.

We choose $\mathcal{C}$ to consist of all 15 corruptions in the IN-C dataset and limit severity to $\mathcal{S} = \{1, 2, 3\}$ in order to further test the sensitivity of the IQA metrics (**D1**). We create variants of the IN-val dataset for $p_c = N/100$ for $N \in [1, \ldots, 20]$. We evaluate the DNNs on these dataset variants and estimate predictability using logistic regression as in previous experiments. We compute $mAUC$ over all $p_c$ variants and find ZSCLIP-IQA ($Q_l$) outperforms all other NR-IQA metrics with $mAUC = 0.64$ with the next best (CLIP-IQA) achieving only $mAUC = 0.57$. The results show that while all metrics can distinguish between differences in the quality distributions of the dataset variants, only ZSCLIP-IQA achieves high predictability over all variants. Conventional IQA metrics improve only as the number of distorted images in the dataset increases (where it becomes easier to separate clean and corrupted images). The full results are found in Appendix H and show that predictability with ZSCLIP-IQA is stable with respect to changes in the proportion of clean/corrupted images in the dataset whereas more traditional NR-IQA metrics remain near random chance $AUC$ and exhibit higher variance as $p_c$ changes.

## 7 DISCUSSION

In this work, we were motivated to identify measures of image quality that allow us to produce IQ-driven priors on DNN performance. We presented a causal inference framework for this problem and proposed a set of IQ metric desiderata to guide our analysis. Using our causal framework, we show conditions where image quality measures can be predictive of DNN performance. We then provide a detailed examination of the relationship between conventional NR-IQA metrics and DNN performance. We use our causal framework and extensive empirical evaluations in the context of image classification to demonstrate that common NR-IQA do not satisfy our desired IQ criteria. We then use the causal approach to develop the task-guided ZSCLIP-IQA metric that provides a causality-driven proof-of-concept metric that satisfies all IQ desiderata and paves the way for future research to improve the alignment between IQA metrics and DNN performance.

**Potential negative societal impacts** As a tool for analysis, the proposed causal framework poses minimal societal risks. While causal models require assumptions about the data generating process, these assumptions are made explicitly in the causal graph and improve the overall transparency of the analysis. Of greater concern is the possibility that using quality metrics to prune/resample datasets may lead to unintended consequences such as removing poor-quality images in a way that disparately affects protected groups. While our work does not address the question of dataset pruning/resampling, we mention this to help ensure that future researchers consider these possibilities in their own work.

**Limitations and future work** We first recognize that image quality alone is insufficient to predict task model performance as both image content and composition play a role in task difficulty and the relationship between IQA and performance may be confounded by other factors. Nonetheless, we show that our causal framework still allows us to analyze the conditions where quality properties of our dataset may be correlated with DNN performance. Our ZSCLIP-IQA method provides one solution that satisfies the proposed IQ desiderata but we believe there are many opportunities for improving on this approach in future research.

We also acknowledge that our experiments only addressed image classification tasks. We focused initially on classification since it is well-studied and clearly defined, with many public benchmarks available for evaluation. Even in this context, we are the first to show that the notion of quality is task-dependent (i.e., perceptual judgement vs. classification). Our primary contribution in this work is the causal framework and we believe this provides a strong foundation for supporting future research that examines similar questions for a wider range of vision tasks.

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
