## A CAUSAL MODEL FOR NR-/FR-IQA

We provide here the causal models for the NR-IQA and FR-IQA settings in Figure 8.

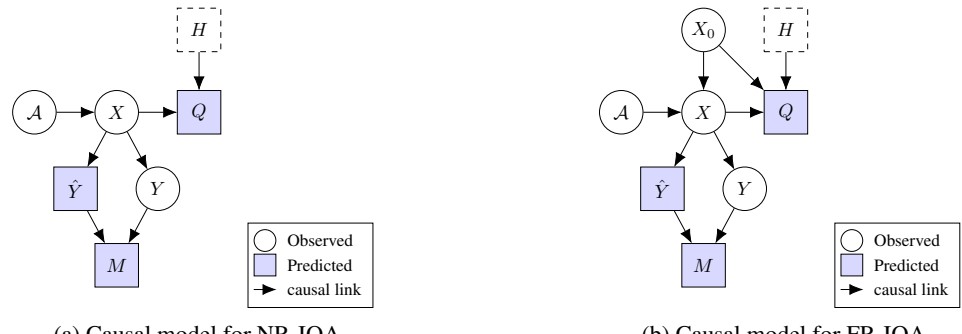

(a) Causal model for NR-IQA  (b) Causal model for FR-IQA

Figure 8: Causal model for the NR-/FR-IQA settings. The FR-IQA setting includes $X_0$ which is the reference image. In both models, $H$ indicates human annotator guidance which reflects that IQ metrics are typically calibrated against human perceptual judgements (dashed-line box indicates $H$ is not used directly in the calculation of $Q$).

While the main manuscript focused on the NR-IQA setting, we can see from the causal models here that the results generalize to the FR-IQA case as well. In particular, the independence of $Q, M$ given $X$ is not affected by whether a "clean" reference image ($X_0$) is available for computing $Q$. Also, these models also account for the influence of human annotators $H$ in calibrating the function for computing $Q$, but not that this does not change the relationship between $M, Q$.

## B CAUSAL MODEL FOR IQA WITH LATENT FEATURES

In understanding the difference between the baseline IQA formulation in Figure 1 and the shared features formulation of Figure 2 in Section 3, we provide an expanded version of the baseline DAG in Figure 9. Here we show that the task DNN for computing $\hat{Y}$ and the function for computing $Q$ rely on latent features $Z_{\hat{Y}}$ and $Z_Q$ respectively.

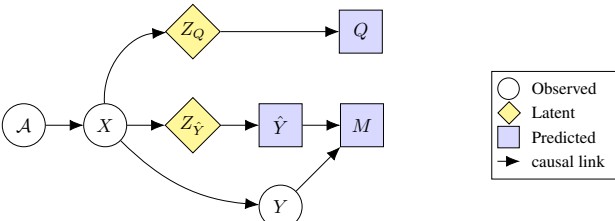

Figure 9: Causal model for IQA that accounts for the use of latent features by the task DNN and IQ metric towards computing $M$ and $Q$ respectively.

In this expanded model, $Z_{\hat{Y}}$ and $Z_Q$ are independent given $X$ and not "shared", and therefore $Q \perp M \mid X$ as discussed in §3. In contrast, Figure 2 considers the case where $Z$ represents the features derived from $X$ that are common between $Z_{\hat{Y}}$ and $Z_Q$ shown in the baseline case above. Thus, Figure 2 shows the case where $X$ does not block all paths between $Q, M$ since a path exists from $Q$ to $M$ through $Z$. This ensures that $Q$ and $M$ will be correlated given $X$ unlike in the baseline case above.

## C CAUSAL MODEL FOR COMMON CORRUPTIONS ROBUSTNESS EVALUATION

The common corruptions framework (Hendrycks & Dietterich, 2019) is used in our experiments to ensure full control of the image distortion types and severity. Figure 10 shows a version of the

baseline IQA causal model customized to account for the corruption process used by this evaluation framework. Here, the corrupted image $X$ is determined by the corruption function (e.g., Gaussian noise, defocus blur, fog, contrast, brightness, JPEG compression), the severity ($S \in \{1, 2, 3, 4, 5\}$), and the "clean" image $X_0$.

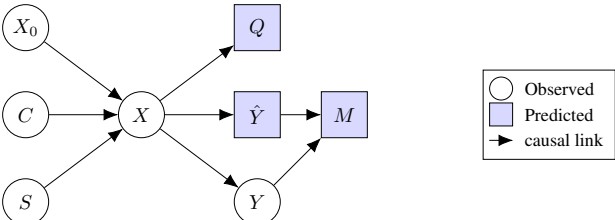

Figure 10: Causal model for the common corruptions framework where $C$ refers to the corruption type, $S$ refers to the corruption severity, and $X_0$ is the unperturbed, "clean" image.

In this setting, we see that $C, S$ replace the original set of imaging factors $\mathcal{A}$ in the graph in Figure 1. As such, the analysis from §3 holds in the common corruptions framework and allows us to study the relationship between $Q, M$ in a setting where we can precisely control the imaging conditions.

## D   RELATIONSHIP OF NR-IQA AND DNN PERFORMANCE METRICS

In Figures 11, 12, and 13 we show the relationship of additional NR-IQA metrics with DNN performance for additional architectures and metrics. In general, we see weak trends in accuracy vs. average IQ suggesting that these metrics are most consistent with the causal model in Figure 1.

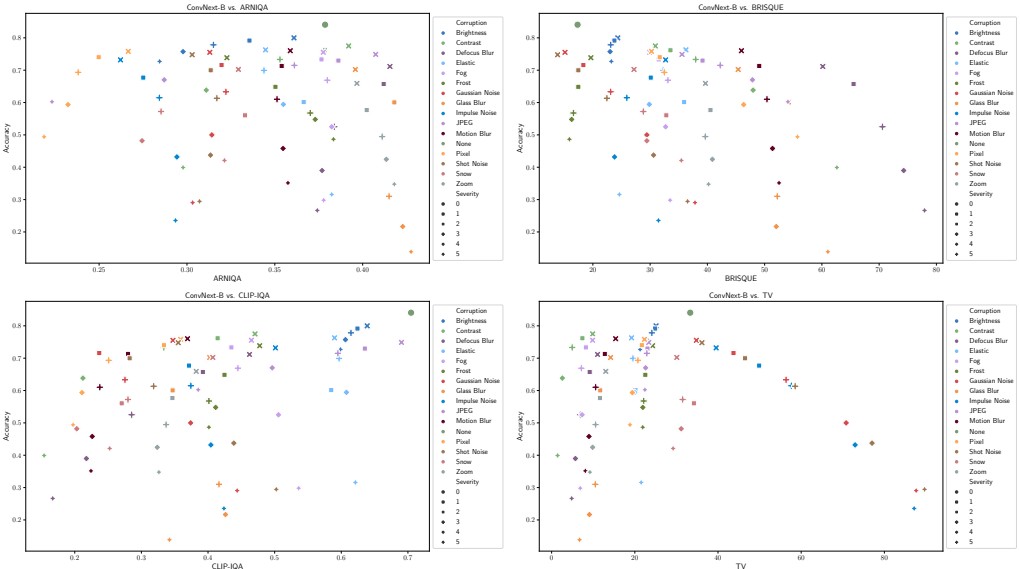

Figure 11: Comparison of ConvNext-B accuracy with (clockwise) ARNIQA, BRISQUE, CLIP-IQA, and TV. Little correlation is observed between group-wise accuracy and each NR-IQA metric.

## E   RELATIONSHIP OF STRONG TASK-GUIDED IQA AND DNN PERFORMANCE METRICS

In Figures 14, 15, 16, we examine the relationship between DNN performance and the strong task-guided metrics ($Q_p, Q_h, Q_l$) described in §5. Each figure pairs the task DNN under consideration with a pre-trained task model used to compute the quality metric.

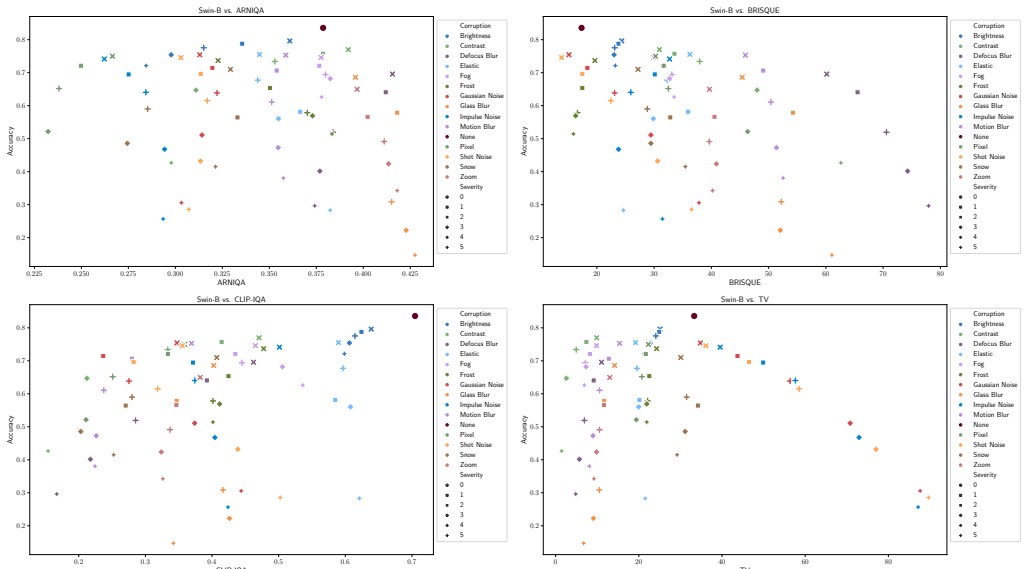

Figure 12: Comparison of Swin-B accuracy with (clockwise) ARNIQA, BRISQUE, CLIP-IQA, and TV. Little correlation is observed between group-wise accuracy and each NR-IQA metric.

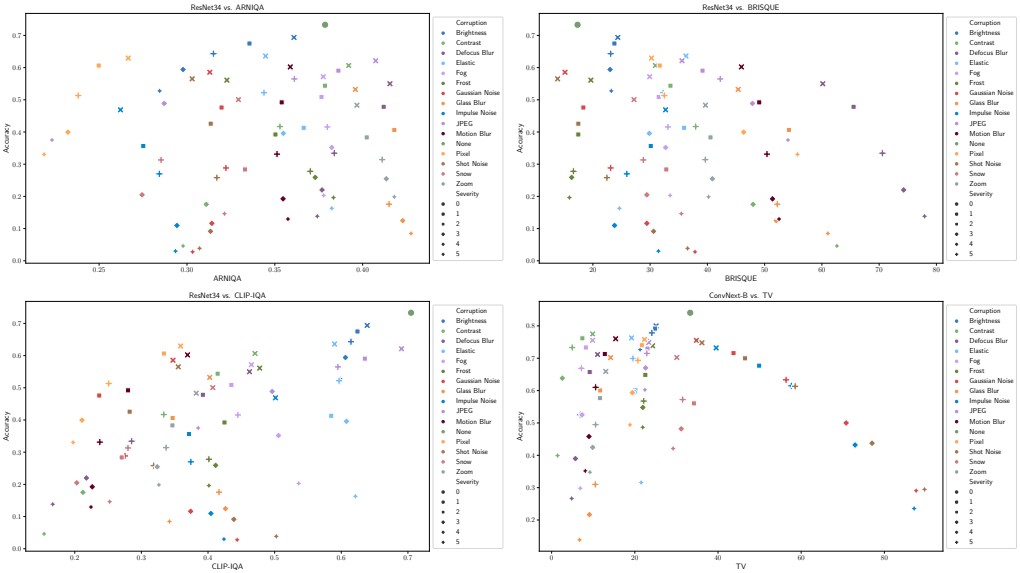

Figure 13: Comparison of ResNet34 accuracy with (clockwise) ARNIQA, BRISQUE, CLIP-IQA, and TV. Little correlation is observed between group-wise accuracy and each NR-IQA metric.

Table 4 also shows the point-wise predictability results for the strong task-guided IQA case. This table extends Table 2 for additional task DNNs. Results here show that strong task-guided IQA metrics are highly correlated with DNN performance and that predictability remains high regardless of whether the pre-trained DNN used to compute $Q$ is the same DNN used to obtain $M$.

# F  RELATIONSHIP OF WEAK TASK-GUIDED IQA AND DNN PERFORMANCE METRICS

We provide Figures 17, 18, 19 showing the relationship between DNN performance the weak task-guided ZSCLIP-IQA metric from §6.

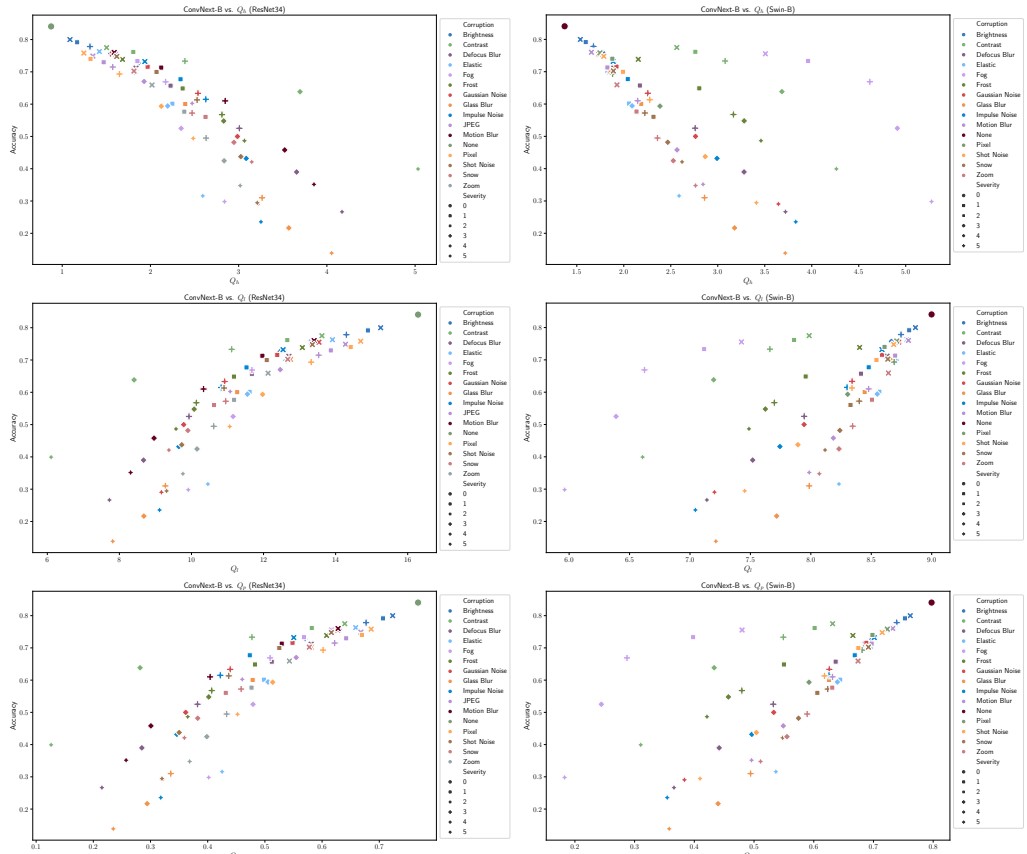

Figure 14: Comparison of ConvNext-B accuracy with (row) $Q_h, Q_l, Q_p$ computed using (col) ResNet34, Swin-B. High correlation is observed between each IQA metric and accuracy.

## G  CONTROLLING FOR IMAGE CONTENT WHEN EVALUATING PREDICTABILITY

The experiments of §4-6 examined predictability by modeling $P(M|Q, X)$. Since the content of $X$ may be a confounder for both $M, Q$, we attempt to control for it in two ways.

In the first case, we take advantage of the synthetic nature of the IN-C dataset by looking at the predictability of $M$ from $Q$ for each image in the dataset separately which allows us to control the content precisely and only change the quality characteristics. For this experiment, we train a logistic regression classifier to predict $P(M|Q)$ for individual image IDs trained using only $M, Q$ computed from the set of distorted variants of each specific image ID. Given the original clean image and 15 corruptions with 5 severity levels each, we run 5-fold cross-validation with an 80/20 train/test split of the 76 total images. We repeat this for all 50k image IDs in the ImageNet validation set. The results shown in Figure 20a are an average of the AUC over all image IDs and folds.

We see that even when controlling for the image content the weak task-guided IQA generally achieves the highest $mAUC$ with the lowest variance. Overall, this supports our hypothesis and causal analysis that weak task-guidance provides a means to associate $M, Q$ even when conditioning on the image directly.

In the second case, we adjust for image content by controlling for the image label $Y$. Here, we train a separate classifier to model $P(M|Q)$ for each of the 1000 labels in the ImageNet dataset. Each classifier is trained on the aggregate of 50 images per label along with all 15 corruptions at 5 severity levels (3751 total per label). We again use an 80/20 train/test split and perform 5-fold cross-validation. The results shown in Figure 20b are an average of the $AUC$ over all labels and folds.

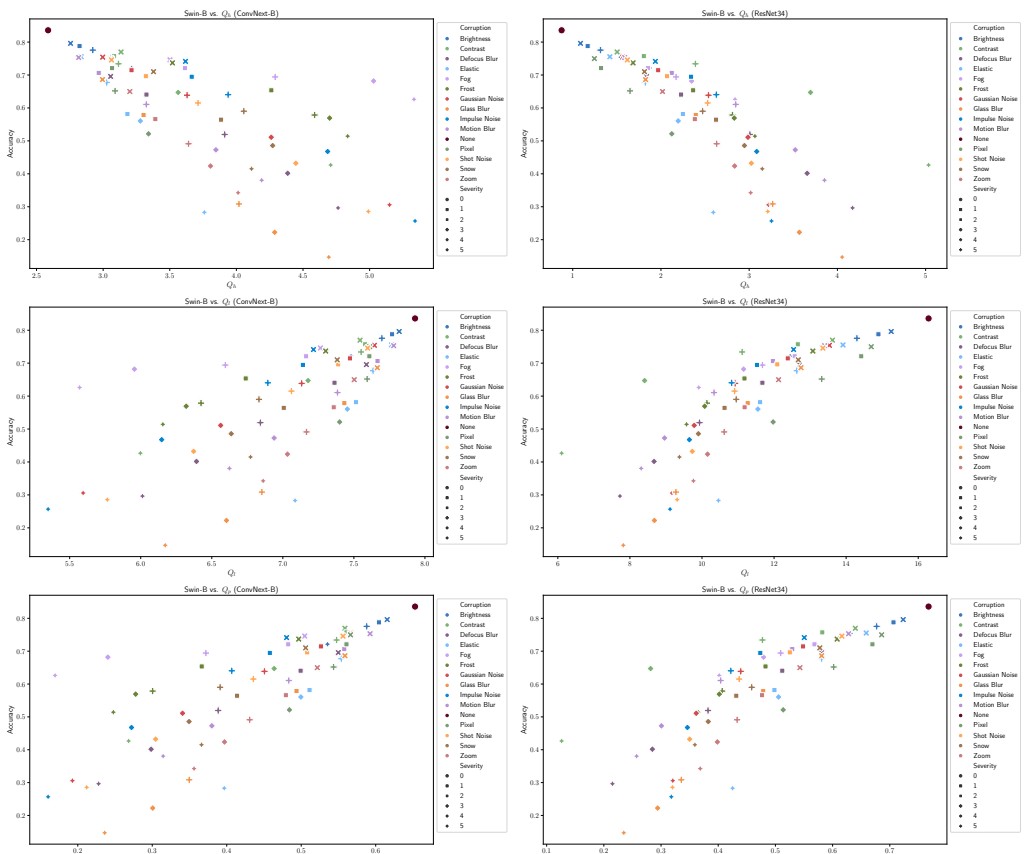

Figure 15: Comparison of Swin-B accuracy with (row) $Q_h, Q_l, Q_p$ computed using (col) ConvNext-B, ResNet34. High correlation is observed between each IQA metric and accuracy.

We find here that across all IQ metrics evaluated, $mAUC$ is barely above chance. For the traditional NR-IQA metrics, this supports our analysis and main experiments which show little correlation between $M, Q$. For the ZSCLIP-IQA metric, we refer again to the causal model (Fig. 6) and see that while the task selection variable ensures the association between $M, Q$, it is the conditioning on $Y$ (and $X$), as done here, that blocks all paths between $M, Q$ and once again removes the association.

# H PREDICTABILITY OF DNN PERFORMANCE FOR MILDLY CORRUPTED DATASETS

To show that $D1$ is satisfied even in the case of mildly corrupted data, we plot the distributions of $Q$ in Figure 21. Across all variants, even in cases where the likelihood is low, each IQA metric exhibits sensitivity to corruption (**D1**).

While some IQA metrics are more sensitive to the overall image corruption, this does not necessarily translate to higher predictability. In fact, ZSCLIP-IQA appears to have smaller differences in IQA distribution across variants compared to other IQA metrics, yet the highest predictability of $M$. Figure 22 shows the predictability of $M$ from $Q$ for variants of the IN-C benchmark created as described in §6.3.

Results show that weak task-guided IQA metrics are able to achieve higher $AUC$ even when the number of corrupted images in the dataset is low. In comparison, conventional NR-IQA metrics achieve lower $AUC$ and are more sensitive to the total level of corruption.

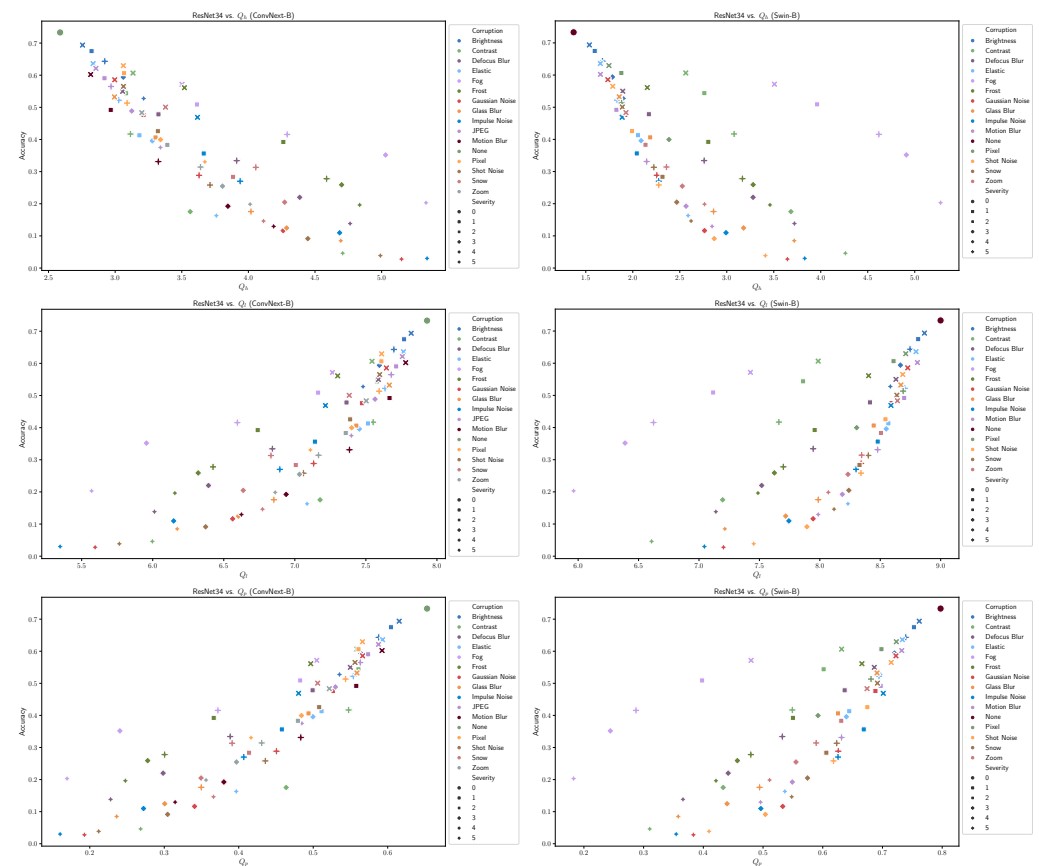

Figure 16: Comparison of ResNet34 accuracy with (row) $Q_h, Q_l, Q_p$ computed using (col) ConvNext-B, Swin-B. High correlation is observed between each IQA metric and accuracy.

## I    COMPUTE RESOURCES

All experiments were run using a single NVIDIA A40 GPU with 48GB of memory. Predictability analysis can be conducted on CPU-only machine with at least 8 cores.

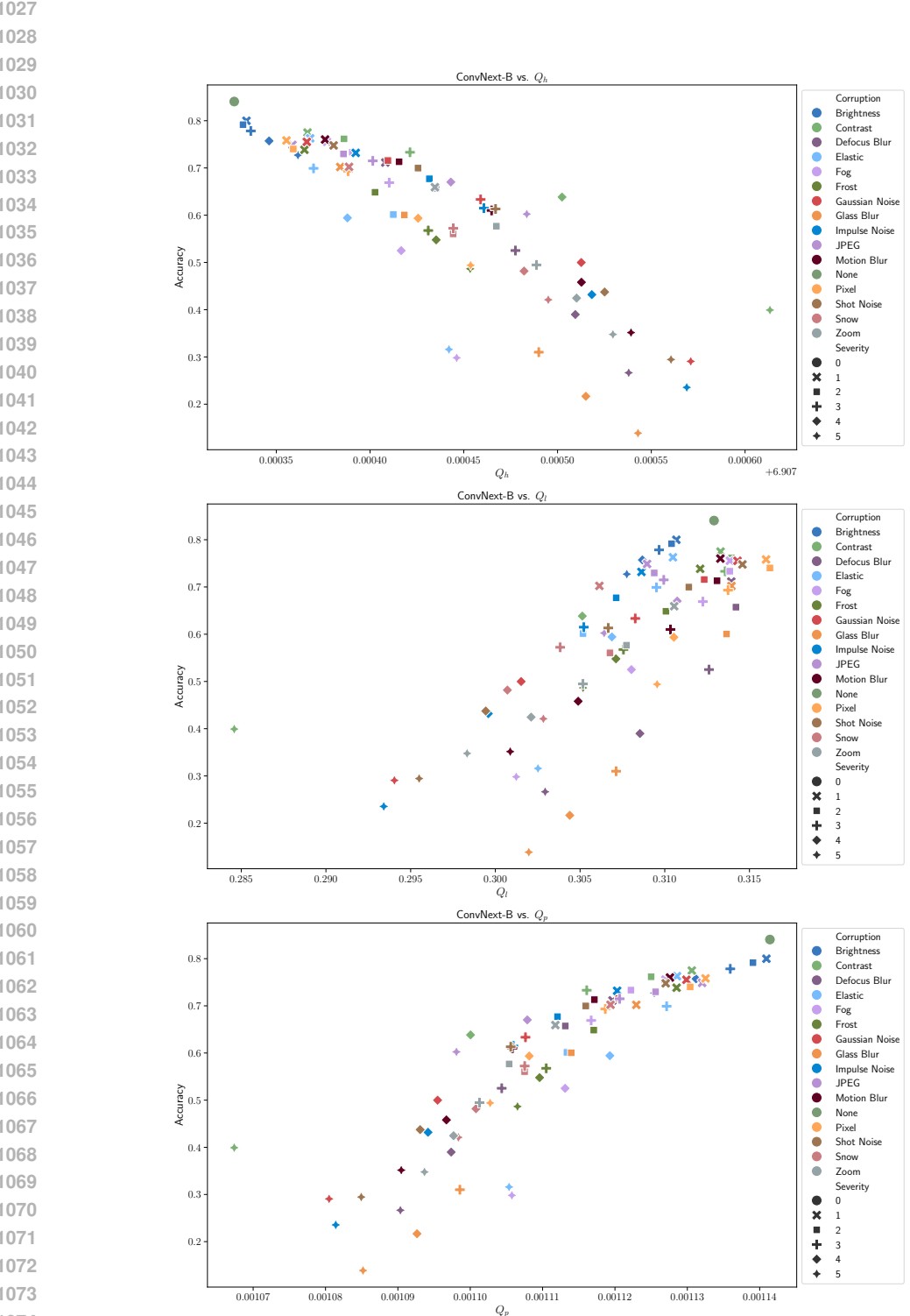

Figure 17: Comparison of ConvNext-B accuracy with (top to bottom) $Q_h, Q_l, Q_p$ based on ZSCLIP-IQA. High correlation is observed between each ZSCLIP-IQA variant and accuracy.

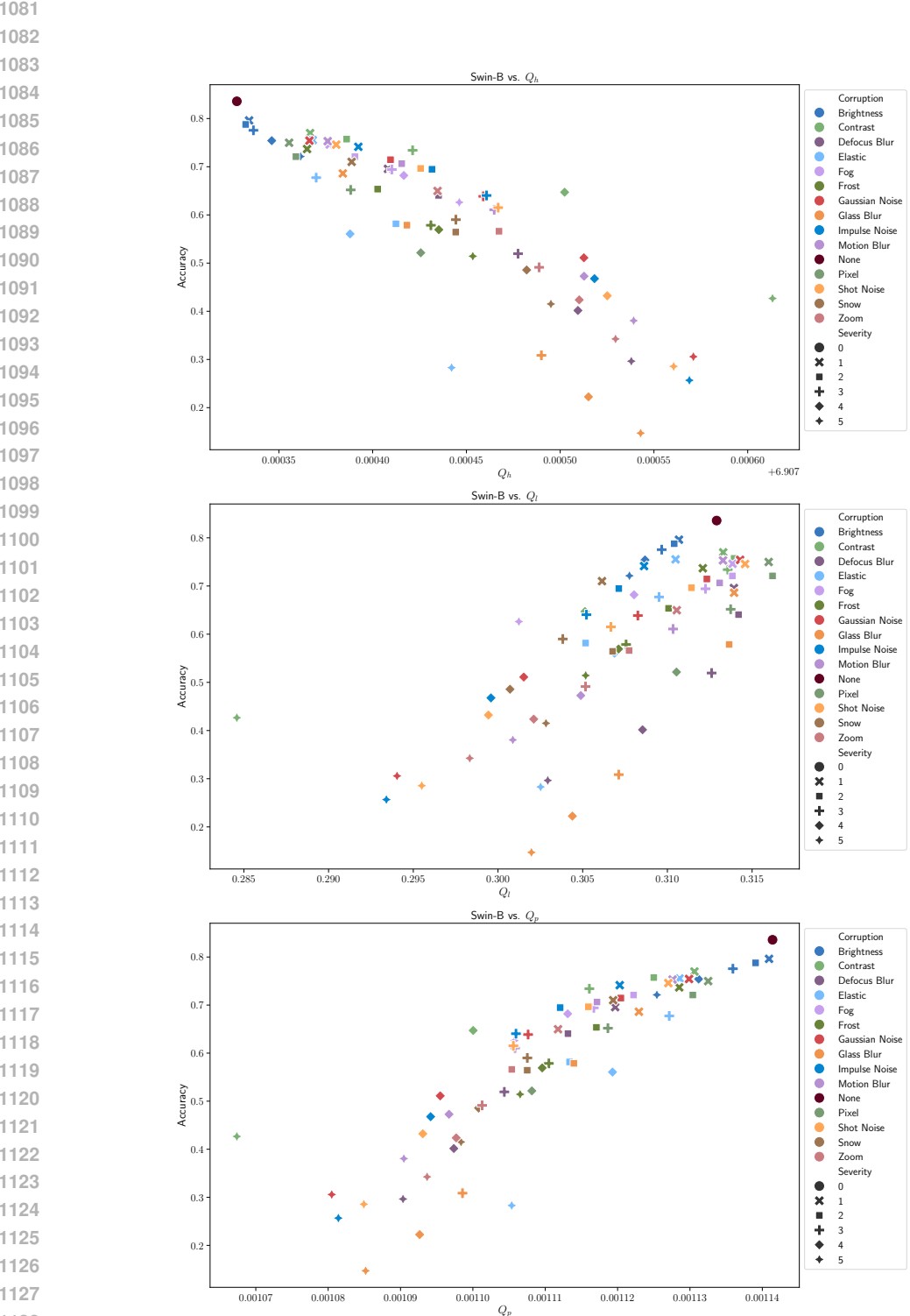

Figure 18: Comparison of Swin-B accuracy with (top to bottom) $Q_h, Q_l, Q_p$ based on ZSCLIP-IQA. Some correlation is observed between each ZSCLIP-IQA variant and accuracy.

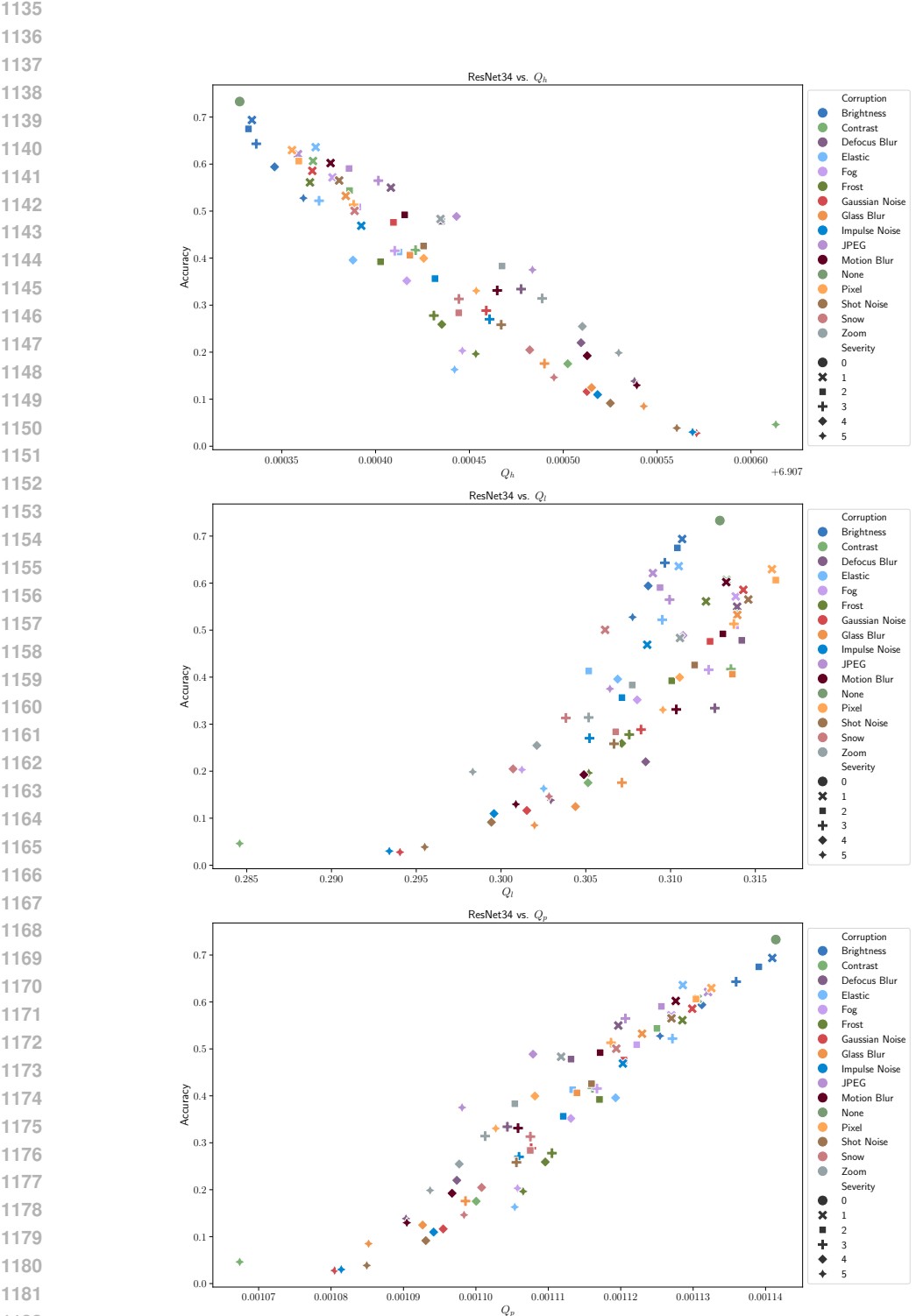

Figure 19: Comparison of ResNet34 accuracy with (top to bottom) $Q_h, Q_l, Q_p$ based on ZSCLIP-IQA. High correlation is observed between each ZSCLIP-IQA variant and accuracy.

Table 4: Correlation between IQ and accuracy. SRCC, PLCC computed using average accuracy for each (corruption, severity). AUC and CE based on point-wise predictions (95% CI within $\pm 0.001$). SRCC, PLCC values have $p < 0.05$.

| Model | IQA Metric | AUC ↑ | CE ↓ | $|PLCC|$ ↑ | $|SRCC|$ ↑ |
|---|---|---|---|---|---|
| ConvNext-B | ConvNext-B $Q_h$ | 0.772 | 0.562 | 0.822±0.070 | 0.854±0.063 |
| | ConvNext-B $Q_l$ | 0.778 | 0.555 | 0.826±0.067 | 0.854±0.063 |
| | ConvNext-B $Q_p$ | 0.826 | 0.504 | 0.888±0.045 | 0.910±0.044 |
| | ResNet34 $Q_h$ | 0.725 | 0.601 | 0.859±0.069 | 0.924±0.045 |
| | ResNet34 $Q_l$ | 0.717 | 0.603 | 0.866±0.051 | 0.925±0.043 |
| | ResNet34 $Q_p$ | 0.719 | 0.601 | 0.875±0.051 | 0.926±0.044 |
| | Swin-B $Q_h$ | 0.760 | 0.579 | 0.624±0.165 | 0.724±0.170 |
| | Swin-B $Q_l$ | 0.724 | 0.601 | 0.604±0.166 | 0.686±0.176 |
| | Swin-B $Q_p$ | 0.791 | 0.547 | 0.742±0.132 | 0.797±0.139 |
| ResNet34 | ConvNext-B $Q_h$ | 0.767 | 0.557 | 0.858±0.060 | 0.889±0.055 |
| | ConvNext-B $Q_l$ | 0.760 | 0.563 | 0.853±0.059 | 0.896±0.055 |
| | ConvNext-B $Q_p$ | 0.801 | 0.522 | 0.904±0.044 | 0.930±0.041 |
| | ResNet34 $Q_h$ | 0.848 | 0.470 | 0.930±0.028 | 0.969±0.023 |
| | ResNet34 $Q_l$ | 0.827 | 0.492 | 0.951±0.015 | 0.973±0.020 |
| | ResNet34 $Q_p$ | 0.850 | 0.461 | 0.960±0.015 | 0.977±0.021 |
| | Swin-B $Q_h$ | 0.751 | 0.574 | 0.643±0.153 | 0.774±0.140 |
| | Swin-B $Q_l$ | 0.709 | 0.600 | 0.628±0.146 | 0.754±0.145 |
| | Swin-B $Q_p$ | 0.774 | 0.551 | 0.747±0.129 | 0.825±0.112 |
| Swin-B | ConvNext-B $Q_h$ | 0.744 | 0.586 | 0.706±0.129 | 0.768±0.098 |
| | ConvNext-B $Q_l$ | 0.746 | 0.586 | 0.709±0.127 | 0.768±0.099 |
| | ConvNext-B $Q_p$ | 0.791 | 0.542 | 0.788±0.102 | 0.834±0.079 |
| | ResNet34 $Q_h$ | 0.722 | 0.603 | 0.828±0.078 | 0.896±0.053 |
| | ResNet34 $Q_l$ | 0.713 | 0.604 | 0.831±0.061 | 0.892±0.053 |
| | ResNet34 $Q_p$ | 0.716 | 0.602 | 0.845±0.062 | 0.897±0.052 |
| | Swin-B $Q_h$ | 0.766 | 0.578 | 0.483±0.207 | 0.654±0.174 |
| | Swin-B $Q_l$ | 0.732 | 0.597 | 0.458±0.203 | 0.611±0.181 |
| | Swin-B $Q_p$ | 0.807 | 0.529 | 0.620±0.184 | 0.732±0.142 |

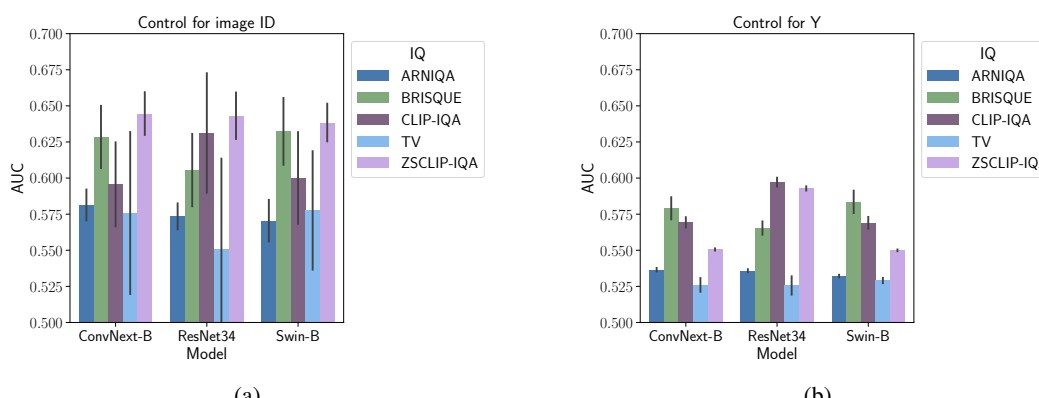

(a)  (b)

Figure 20: Mean AUC (mAUC) for classifiers trained (a) per image ID ($X_i$) to model $P(M|Q, X_i)$ and (b) per label $Y$ to model $P(M|Q, Y)$. Averages are taken over all images/labels respectively and cross-validation folds with error bars indicating one standard deviation. Higher variance in (a) is attributed to lower sample sizes for training the logistic regression classifier.

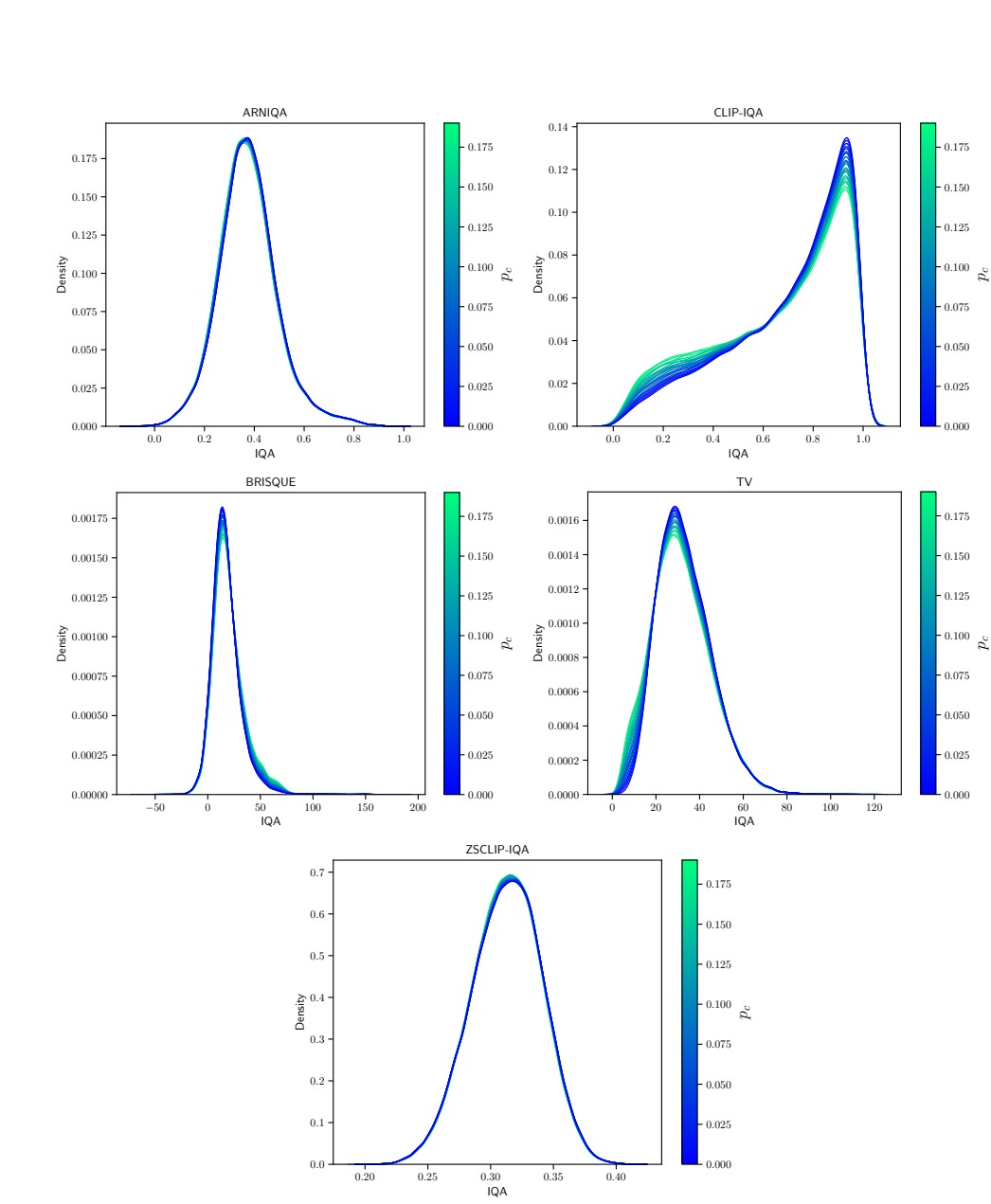

Figure 21: Distribution of IQA for each mildly corrupted variant of IN-val. Line color indicates the likelihood of image corruption $p_c$ for each variant. Note that the amount of similarity/difference in the IQ distribution across variants does not explain the predictability which is determined by the causal DAG such as in §3, §5, and §6. See Figure 22 for predictability results.

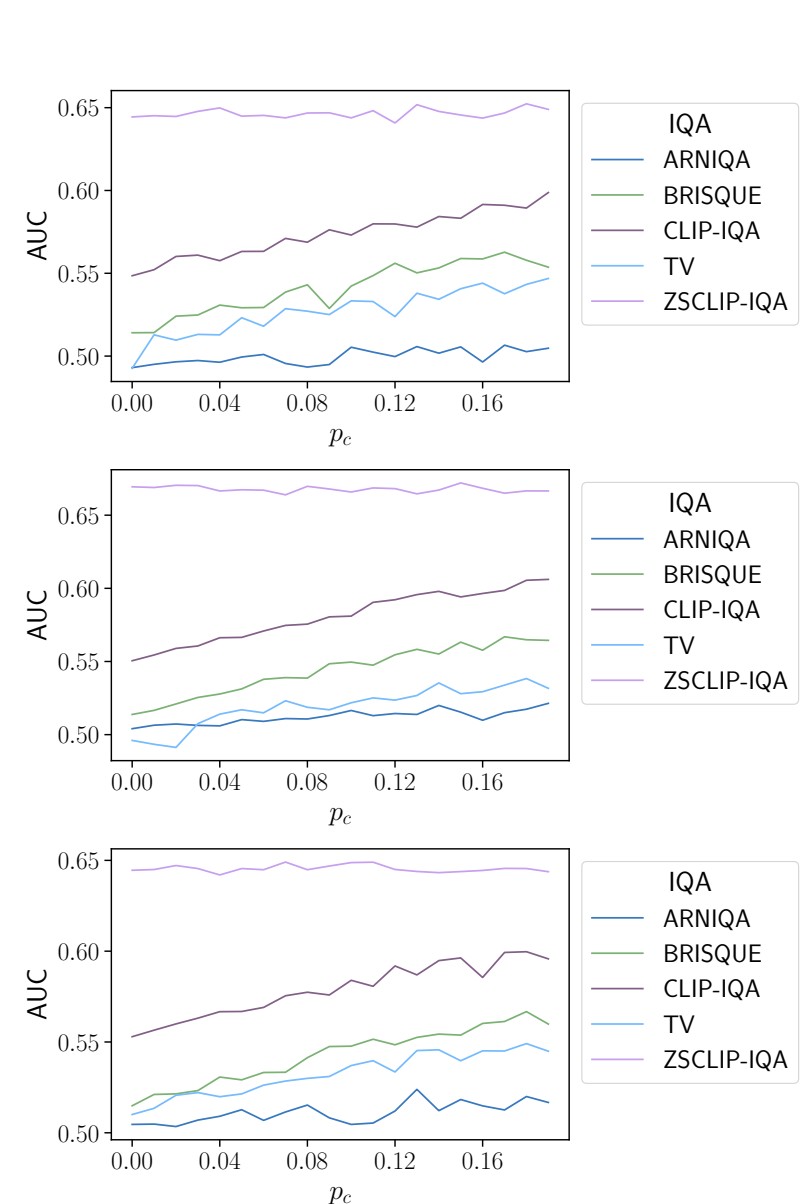

Figure 22: AUC vs. $p_c$ where $p_c$ represents the fraction of images in the test set that are mildly corrupted. Results are listed top to bottom: ConvNet-B, ResNet34, Swin-B. Predictability for ZSCLIP-IQA is relatively insensitive to the proportion of corrupted images whereas other metrics only improve as the proportion and diversity of corruptions increases.