# OpenReview forum: "A Causal Framework for Aligning Metrics of Image Quality and Deep Neural Network Robustness"
_ICLR.cc/2025/Conference — Submitted to ICLR 2025_

### Official Review · Reviewer_AsJJ · 2024-11-03

**Soundness:** 3
**Presentation:** 2
**Contribution:** 2
**Rating:** 5
**Confidence:** 4

**Summary:**

The paper aims to establish a causal link between image quality metrics and DNN performance to enhance robustness. However, the study lacks depth in its analysis and fails to convincingly demonstrate the practical implications of its proposed causal framework.

**Strengths:**

1. The paper aims to bridge the gap between image quality assessment and DNN performance.
2. Theoretical exploration of the relationship between IQ metrics and DNN behavior.
3. The authors provide a new image quality metric aligned with DNN sensitivities.

**Weaknesses:**

* It is unclear why the authors chose to use a causal framework to analyze the relationship between image quality and DNN performance across various IQA settings.
* The notation is overly complex, making it difficult to recognize the technical details of the proposed causal framework.
* The paper does not clarify the differences between Fig. 1 and Fig. 4, nor does it address potential confounders in establishing causal relationships between IQ metrics and DNN performance.
* The theoretical underpinnings of the causal framework are not sufficiently articulated, which may lead to misunderstandings about how the proposed model operates.

**Questions:**

* While the authors emphasize that D4 should be independent of any task-specific model, the TG-IQA is trained for a specific task. This issue requires further elaboration.
* The experiments rely heavily on a single dataset (ImageNet) and do not assess the generalizability of the findings across diverse datasets or tasks.
* The reported correlations between the proposed IQ metric and DNN performance are weak. A more detailed analysis of these results, including confidence intervals and significance levels, is necessary.
* The paper suggests that the new IQA metric can serve as a prior for DNN performance without adequately demonstrating this capability in a variety of contexts, such as the IQA performance validation.

---

> ### Author Response · Authors · 2024-11-14
> **Author response (1/2)**
>
> Thank you for your time and thoughtful feedback.  We will use these concerns to continue to improve our manuscript and we're glad to further discuss any of the points of clarification below.
>
> > It is unclear why the authors chose to use a causal framework to analyze the relationship between image quality and DNN performance across various IQA settings.
>
> While not the only approach to answering these questions, we used the causal approach for the following reasons:
> 1. It gives a means to state our explicit assumptions about the key variables influencing both $Q, M$ and their expected interactions
> 2. It enables us to easily compare different perspectives on the same problem (e.g., Figures 1, 2, 4, 6)
> 3. It provides mathematical tools for understanding conditions for and differences between causation and association
> 4. We can expand upon this framework in future work to understand how the data generating process (e.g., via $\mathcal{A}$) or task (e.g., via the nature of $\mathcal{Y}$) may yield insight into how IQ metrics can be more informative of performance metrics.
>
>
> > The notation is overly complex, making it difficult to recognize the technical details of the proposed causal framework.
>
> Thank you for raising this point and we will revisit our current choice of notation, especially by leading with a more easily accessible verbal introduction to provide intuition before delving more deeply into mathematical fomulation. We are happy to simplify further if there are particular suggestions the reviewer might offer.
>
>
> > The paper does not clarify the differences between Fig. 1 and Fig. 4...
>
> In Fig. 4, $Q$ is computed from $\hat{Y}$ (i.e., there is an edge $\hat{Y} \rightarrow Q$) whereas in Fig. 1, $\hat{Y}$ and $Q$ are conditionally independent given the image $X$.  We will clarify this directly in Section 5.
>
> > ...nor does it address potential confounders in establishing causal relationships between IQ metrics and DNN performance.
>
> This is a valid question, so let's first start with our current framework.  Since $\hat{Y}$ is deterministically computed from $X$ using the task DNN and $M$ is deterministically computed from predictions $\hat{Y}$ and labels $Y$, we can rule out hidden confounders between $X$ and $M$.  Similarly, in the most generic setting (Fig. 1), the quality score $Q$ is computed directly from $X$ using the IQ metric so we can again reasonably assume no hidden confounders.  As such, we can assume causal sufficiency in the setting that we've described.
>
> In a more general sense, any common cause of $Q$ and $M$ could be considered a confounder, so excluding the images $X$ and labels $Y$ themselves, any factor captured in $\mathcal{A}$ might also be informative about $Q, M$.  For instance, in ImageNet-C (see Appendix C), this amounts to knowing the corruption $C$ and severity $S$ which both directly impact the image $X$. Given our causal framework and the model of Figure 10, we could exploit this knowledge directly to design a quality metric that simply scores images based on $C, S$ (without even observing $X$). This example obviously circumvents the purpose of measuring quality given images alone, but does show how knowledge about common causes of the the image, quality metric, and performance metric could be used to better explain the presence/absence of correlation between $Q$ and $M$.
>
> If the reviewer has other specific potential confounders in mind, we would also be interested to hear and discuss them.
>
>
> > The theoretical underpinnings of the causal framework are not sufficiently articulated, which may lead to misunderstandings about how the proposed model operates.
>
> Thanks for raising this point.  We may be too close to the work and could use a bit more specific guidance about what additional portions of the methods (Section 3?) might remain unclear.

---

> ### Author Response · Authors · 2024-11-14
> **Author response (2/2)**
>
> > While the authors emphasize that D4 should be independent of any task-specific model, the TG-IQA is trained for a specific task. This issue requires further elaboration.
>
> This concern is well-taken. The D4 criterion was intended to encourage quality metrics that could be predictive of task DNN performance without being biased by the specific implementation details of the task DNN.  In Section 5, D4 is violated because we use, e.g., the max logit from a task DNN for $Q$ but also the same task DNN for computing $M$.  In this way, $Q$ is biased to be more predictive of $M$ for the specific task DNN backbone used.
>
> To alleviate this concern, we used the ZSCLIP-IQA approach to show that we could find a way to still rely on task information without requiring the use of a backbone trained on the same data for the exact same task.  While the CLIP backbone for ZSCLIP-IQA was still pre-trained on a similar task and image domain, the authors of CLIP ensured that the CLIP model never saw the ImageNet dataset nor was trained explicitly for the ImageNet labels.  As such, because ZSCLIP-IQA is also a zero-shot method, we can be confident that the zero-shot predictions are not biased by particular design choices of task DNNs for the ImageNet task.  This was also discussed starting at Line 468, but please let us know how we can strengthen that discussion.
>
>
> > The experiments rely heavily on a single dataset (ImageNet) and do not assess the generalizability of the findings across diverse datasets or tasks.
>
> We also recognize this as a limitation of our work, and offer the following points for consideration and context:
>
> 1. Our primary objective was to establish if there was a relationship between IQA metrics calibrated to human perceptual judgements and DNN performance.  We relied on image classification because it has been extensively studied and well-defined, especially in the non-adversarial robustness context.  Other vision tasks require more complex algorithmic solutions where that complexity made it more complicated to precisely study the IQA/performance relationship.
> 2. Our causal framework makes no assumptions on the types of vision tasks or labels, and in theory, is expected to generalize.
> 3. We opted for ImageNet in particular because it is an established benchmark, it included the also established ImageNet-C variant that enabled precise control of image quality, and it sufficiently aligns with natural image domains in which traditional IQA metrics are developed. Other image classification benchmarks fail to address all three of the aforementioned properties.
>
>
> > The reported correlations between the proposed IQ metric and DNN performance are weak. A more detailed analysis of these results, including confidence intervals and significance levels, is necessary.
>
> While the correlations are weak for traditional IQ metrics (Table 1), the correlations and predictability for both strong/weak TG-IQA methods are considerably stronger (Table 2-3).  Generally, cince $Q$ is only measuring the quality aspect of the images, we expect that they won't be fully predictive but that they can be informative priors on DNN performance (Lines 60-62).
>
> Also, confidence intervals and significance are provided in all Tables (or indicated in the caption).  We can provide additional analysis in each experiment section such as observations about differences in performance between task-guided metric variants or relative strengths of predictability across task DNN architectures.
>
>
> > The paper suggests that the new IQA metric can serve as a prior for DNN performance without adequately demonstrating this capability in a variety of contexts, such as the IQA performance validation.
>
> We attempted to show the application of our approach to obtaining DNN performance priors in Section 6.3.  This experiment directly tests the predictability of IQ metrics across a variety of datasets with varying levels image quality. The results show that ZSCLIP-IQA provides a quality measure that remains predictive of DNN performance across all dataset variants (see Fig. 22). The fact that the predictability (AUC) is maintained is direct evidence that ZSCLIP-IQA could be used as a prior for DNN performance since observing the quality score gives information about the overall difficulty for the task DNN.
>
> This result shows the value of our causal framework in enabling the design of new metrics like ZSCLIP-IQA, but we will be clear in the manuscript that we are not making a broader claim of the generalizability of the specific ZSCLIP-IQA metric.  Studying ZSCLIP-IQA (and other future metrics developed with our framework) in a broader variety of contexts is fertile ground for future research beyond this current submission.

---

### Official Review · Reviewer_KDbi · 2024-11-03

**Soundness:** 3
**Presentation:** 2
**Contribution:** 3
**Rating:** 5
**Confidence:** 3

**Summary:**

This paper seeks to investigate the effect of training image quality on the performance and robustness of trained DNN outcomes. They first characterize the inefficiency of conventional IQA methods targeting HVS under the classification context and then propose a causal framework to formulate another image quality metric correlating to DNN performance.

**Strengths:**

It is a very interesting and fresh view to investigate the causal relations between quality-related features and task-related features in DNNs. The problem formulation and solution are clearly stated.

**Weaknesses:**

Despite the efforts mentioned above, there remain several weaknesses in this paper, as stated below.
* The wording and phrasing of this paper are intricate, which significantly impacts its readability.  The text is too long and not coherent, and there are numerous grammatical errors throughout, making it somewhat difficult to understand.
* In this work, the authors restrict the definition of _image quality_ to a very narrow scope, which is the authentic distortions. However, in many mainstream data augmentation methods in the image classification task, applying artificial distortions to original images is a common practice. What is the reason that the proposed framework cannot be extended to artificially distorted image quality?
* In contrast to the previous comment, the authors utilized artificially distorted images, such as JPEG compression and additional noise, as the corrupted images. Is this a contradiction to the previous statement?
* In line 141, what is _priori_ knowledge?
* In the introduction, the authors mentioned not only the concept of image quality but also the image difficulty regarding content. However, I cannot find relevant measures on this aspect aside from the _technical quality_ indicators such as corruption, contrast, or capture noise, while the content distribution of training datasets is of crucial importance for image classification tasks. Is this an implicit assumption that the dataset is balanced in content?
* In line 195, how to understand _at least_ correlated? If the relations between X, Q, and M cannot support strong causation, then how could the entire causal presumption stand?
* In sec. 4.1, the authors employ IN-val and IN-c as clean and corrupted image sets, respectively. Why can you regard the IN-val as _clean_ under the NR-IQA context?
* The contribution of this paper reminds the reviewer of _core set selection_.  Can this paper contribute to this aspect?
* This paper primarily assumes without substantial justification that image quality metrics directly correlate with DNN robustness or performance under varying conditions. This assumption might be overly simplistic.

**Questions:**

See my comments above.

---

> ### Author Response · Authors · 2024-11-14
> **Author response (1/2)**
>
> We thank the reviewer for their feedback and their appreciation of the clarity and quality of our framework and results.  We welcome further discussion on any of the points below.
>
> > The wording and phrasing of this paper are intricate, which significantly impacts its readability. The text is too long and not coherent, and there are numerous grammatical errors throughout, making it somewhat difficult to understand.
>
> Thank you for flagging this issue. We will simplify the text and correct grammatical errors, and would welcome recommendations on which sections in particular are hard to follow, for us to focus our attention on those.
>
>
> > What is the reason that the proposed framework cannot be extended to artificially distorted image quality?
> > The authors utilized artificially distorted images, such as JPEG compression and additional noise, as the corrupted images. Is this a contradiction to the previous statement?
>
> This is a good question and one we can address here and in the manuscript.  In short, our framework should extend to both authentic and artificial distortions as the reviewer has observed. This is captured in cases like Figures 1, 2, 4, and 6 where the $\mathcal{A}$ node represents the imaging factors that influence the appearance of images $X$.  The question about the relationship between $Q$ and $M$ is really driven by whether they are sensitive to the same types of image features or not (i.e., $Z$ in Fig. 2). In that sense, one could use our causal framework analyze the relationship between traditional IQA metrics and DNN performance for any distribution of distortions caused by factors $\mathcal{A}$ (authentic or synthetic).
>
>
> > In line 141, what is a priori knowledge?
>
> This simply means that we don't want our IQ metric to be designed or biased based on knowledge about what task DNNs might be trained/evaluated on the data. In other words, we don't want the IQ metric to be designed to preferentially favor one task DNN architecture over another (e.g., Swin-B vs. ConvNext-B) or one task model pre-training dataset over another (e.g., ImageNet vs. LAION-5B).  Instead, we wish to use the IQ metric to measure the quality of images in the target dataset while making no assumptions about how that data will be consumed. We will clarify this in the manuscript.
>
>
> > In the introduction, the authors mentioned not only the concept of image quality but also the image difficulty regarding content...Is this an implicit assumption that the dataset is balanced in content?
>
> This is a great question and one that we will clarify in Section 1.  We do make the assumption for this work that the dataset is balanced in content.  Because of the careful data collection and annotation process used in building large-scale datasets like ImageNet, we feel this is a reasonable assumption.  The reviewer makes a good point that this is an important aspect of understanding DNN performance and we think that there is room for future work in this area of separating content and appearance.  We discuss some of the current work in characterizing difficulty in Section 2, but will try to make this connection more explicit in the final manuscript.
>
>
> > In line 195, how to understand "at least correlated"? If the relations between X, Q, and M cannot support strong causation, then how could the entire causal presumption stand?
>
> There is an important distinction that we aim to clarify here and in Section 3.3.  The causal model itself does capture causal relationships that we do understand, such as arrows from $X$ to $Y$ that represent the relationship between an annotator reviewing the image and assigning a label or arrows from $\hat{Y}, Y$ to $M$ that represents how the performance metric is calculated.  However, we know that there is no causal relationship between $Q, M$ (i.e., no arrow between them) because we neither compute image quality using the task DNN performance metric nor do we compute the performance metric using the quality score.
>
> However, even though the causal link does not exist between $Q, M$, we can use the causal model to help us determine whether there might still be a correlation between $Q, M$.  In the context of estimating the quality distribution of a dataset, this correlation may be sufficient since, in the correlated case, observing the distribution of $Q$ values will still give us information about the dataset even before we've computed $M$.  So our causal framework shows that while $Q, M$ may not be causally related, the framework can be used to identify the conditions when they might *at least* be correlated.
>
> Please let us know if we can further clarify this point.

---

> ### Author Response · Authors · 2024-11-14
> **Author response (2/2)**
>
> > In sec. 4.1, the authors employ IN-val and IN-c as clean and corrupted image sets, respectively. Why can you regard the IN-val as clean under the NR-IQA context?
>
> This view of IN-val is adopted from traditional robustness research since we know that no corruptions have been applied to the original images in IN-val (as compared to IN-C).
>
>
> > The contribution of this paper reminds the reviewer of core set selection. Can this paper contribute to this aspect?
>
> This is an interesting connection that we think could be related to our work here (although we are not experts in core set selection methods).  For instance, perhaps we could use the task performance-aligned IQ metrics identified by our framework to identify subsets of the dataset that collectively represent the full distribution (from a quality perspective) and that when used for training they will yield task models with similar robustness to those trained on the full dataset. We can add a note about this connection to related work and find it an interesting avenue for future research.
>
>
> > This paper primarily assumes without substantial justification that image quality metrics directly correlate with DNN robustness or performance under varying conditions. This assumption might be overly simplistic.
>
> We clarify that addressing this exact question is really the primary aim of this work (see Lines 69-70). This question has not been studied in depth (if at all) prior to this work and our work provides an initial answer to the question both through our causal framework and through empirical results that show traditional IQ metrics do not correlate with DNN robustness.  We hope that our work will enable other researchers to more deeply examine this question for a broader set of vision tasks and image datasets.

---

### Official Review · Reviewer_qL7W · 2024-11-03

**Soundness:** 3
**Presentation:** 4
**Contribution:** 3
**Rating:** 8
**Confidence:** 4

**Summary:**

The paper proposes a causal framework to determine the relationship between IQA metrics and DNN robustness. The authors run a study with two recent DNN-based IQA methods on how these metrics predict DNN robustness, finding them weakly predictive. The work ends with developing a task-guided IQA metric that aligns better with DNN sensitivities.

**Strengths:**

1. First, I appreciate how well the authors have presented the paper. The paper is relatively easy to read, and the results are nice to interpret.
2. A causal framework helped set up the problem of predicting DNN robustness using IQA metrics. Also, clearly stating the desired outcome along with the problem formulation was helpful.
3. A well-enough coverage of IQA metrics (ranging from BRISQUE to CLIP-IQA).
4. Convincing empirical results to support the theoretical framework.

**Weaknesses:**

1. Experiments limited to the Image Classification task would have been preferred if Object detection/ Image segmentation tasks could have been added to strengthen the results of the proposed causal framework substantially.
2. (minor enhancement) Figure 3 (and similar figures), it is difficult to assess the correlations per distortion category from a single plot . This is especially true in the case of Figure 3, where the correlation is poor. Can a better way to represent visually be provided? Apart from the results reported in the tables.
3. Domain Gap:   The experiments are conducted on ImageNet and its corrupted synthetically generated variant, which might not represent the diversity of real-world data.
4. Only one image classification dataset is used. It would be interesting to see results on images from one of  SVHN (http://ufldl.stanford.edu/housenumbers/), CIFAR-10/100  (https://www.cs.toronto.edu/~kriz/cifar.html), MNIST (https://yann.lecun.com/exdb/mnist/)

**Questions:**

1. Can the authors propose a solution based on the causal framework that can be unimodal? For example, the Image Classification task is unimodal, leveraging multimodal representations to solve the problem while being acceptable. It would be interesting to explore a solution using only image domain representations.
2. While the results with ImageNet are convincing, could the authors run experiments with at least one other image classification dataset, such as CIFAR10/100, SVHN, or MNIST? Or at least provide some explanation.

---

> ### Author Response · Authors · 2024-11-14
> **Author response (1/2)**
>
> We thank the reviewer for their time, their thoughtful feedback, and their insightful questions.  We look forward to discussion of any of the points made below.
>
> > Experiments limited to the Image Classification task would have been preferred if Object detection/ Image segmentation tasks could have been added to strengthen the results of the proposed causal framework substantially.
>
> We agree that considering other tasks would strengthen the paper. That said, we feel that focusing only on image classification does not diminish the impact of our results for a few reasons:
>
> 1. Image classification has been widely studied especially in the non-adversarial robustness context, and so establishing for the first time that traditional IQA metrics are not predictive of DNN performance even just for this task implicitly has broad impact.
> 2. Our causal framework does not make any assumptions about the task or labels, so the theoretical results hold broadly even if the experiments are narrowly focused.
> 3. While detection/segmentation are well-defined problems with existing benchmarks, the algorithmic solutions to these tasks are more complex. This added complexity was expected to confound interpretations of relationships between traditional IQA metrics and performance, so we opted to focus deeply on image classification so we could effectively address our research question (Lines 69-70).
>
>
> >(minor enhancement) It is difficult to assess the correlations per distortion category from a single plot . This is especially true in the case of Figure 3, where the correlation is poor. Can a better way to represent visually be provided? Apart from the results reported in the tables.
>
> We will consider other means of conveying these results.  Would including a (dashed) line for the optimum correlation help to show how far the trend deviates from the desired result?
>
>
> > Domain Gap: The experiments are conducted on ImageNet and its corrupted synthetically generated variant, which might not represent the diversity of real-world data.
>
> We agree that this is a concern, but opted for this route primarily for two reasons which we will clarify in the manuscript:
>
> 1. For evaluation purposes, we needed to clearly control and understand how imaging conditions differed and thus opted to use the corruptions of ImageNet-C since they address these needs.
> 2. Existing "in-the-wild" datasets may offer more diverse, real-world conditions, but either they don't have task labels (e.g., traditional IQA benchmarks) or the conditions can't be verified (without resorting to human judgement).
>
> In fact this question of assessing quality in real-world datasets was a primary motivation for this work.  We originally wished to use IQA metrics to assess quality in real-world data but uncertainty about whether traditional NR-IQA metrics were informative of DNN performance led to the development of our causal framework to answer this question.
>
>
> > W4: Only one image classification dataset is used.
> > Q2: While the results with ImageNet are convincing, could the authors run experiments with at least one other image classification dataset, such as CIFAR10/100, SVHN, or MNIST? Or at least provide some explanation
>
> We considered other common benchmark datasets for image classification, but ran into the following problems (which we can discuss in the manuscript as well):
>
> 1. Quality metrics for thumbnail images like in CIFAR10/100, SVHN may not be particularly informative since the low resolution is likely to obscure quality properties (unless the distortion is already significant).
> 2. MNIST images are essentially uniform in "quality", so we do not expect IQA metrics to provide any information in that setting.
> 3. Aside from the image resolution issue, SVHN does have some variability in quality, but applying the ImageNet-C corruptions to this dataset likely would not have offered more information than evaluation with ImageNet(-C) directly.
>
> We are open to considering additional image datasets for classification and will assess whether we can run training/evaluation to generate additional Tables 1-3 for another benchmark.

---

> > ### Comment · Reviewer_qL7W · 2024-11-25
> >
> > Thanks authors for the reply.
> >
> > Few points :
> > 1. Please discuss why datasets like CIFAR10/100, SVHN were not used in the manuscript, as mentioned in the reubttal.
> > 2. Please improve Fig 3 in the paper. A dashed line might help, but look forward to better methods.

---

> ### Author Response · Authors · 2024-11-14
> **Author response (2/2)**
>
> > Can the authors propose a solution based on the causal framework that can be unimodal? For example, the Image Classification task is unimodal, leveraging multimodal representations to solve the problem while being acceptable. It would be interesting to explore a solution using only image domain representations.
>
> This is an interesting question and if we understand correctly, the question is about whether we can use the causal framework to satisfy all desiderata by relying solely on image representations (i.e., not text-based task conditioning as with ZSCLIP-IQA).  This idea is best captured by Figure 2 which shows that in the case that $Q$ and $M$ both rely on a common set of image-features, then we can be confident there will be some association between them.
>
> Akin to the strong TG-IQA setting, prior knowledge about the task DNN or imaging domain might be used to identify a DNN backbone for $Q$ that is trained on a similar but distinct imaging domain (to satisfy D4) that can produce image embeddings/quality scores that correlate well with image quality and task performance (D1-D3).  We can add some discussion about this idea to the manuscript to show how the causal framework enables a way of thinking about the problem and solution space.
>
> Can the reviewer clarify if we're understanding their question correctly?

---

> > ### Comment · Reviewer_qL7W · 2024-11-25
> >
> > Yes, it would be good to have some discussion on these lines.
> >
> > Thanks for the reply.

---

### Official Review · Reviewer_xKhd · 2024-11-04

**Soundness:** 2
**Presentation:** 3
**Contribution:** 2
**Rating:** 3
**Confidence:** 2

**Summary:**

This work investigates how image quality impacts the performance of deep neural networks (DNNs). The authors propose that aligning image quality metrics with DNN sensitivities could allow these metrics to serve as proxies for predicting image or dataset difficulty. Using a causal framework, the authors examine the predictive power of IQA metrics on DNN classification accuracy and find them to be weak indicators. Building on these findings, the authors introduce a new image quality metric that better correlates with DNN performance. This novel metric offers a more effective method for estimating the quality distribution in large image datasets.

**Strengths:**

The authors proposed an generalised notation for IQA metrics and how it’s connected to classification accuracy. The paper is well-organised, well-written and easy to follow.

**Weaknesses:**

The main weakness is that no comparison of ZSClip-iqa with recognition-aware quality metrics is given. There is a class of IQA metrics that predict not subjective quality, but classification accuracy. For example, “ Towards Machine Perception Aware Image Quality Assessment”, “ Quality assessment for face recognition based on deep learning”, “ Ser-fiq: Unsupervised estimation of face image quality based on stochastic embedding robustness.”

**Questions:**

1) What is the relation of the proposed metric and the existing IQA metrics designed for face/image classification accuracy?
2) What is the transferability of the proposed metric for predicting DNN performance in tasks other than classification?

---

> ### Author Response · Authors · 2024-11-14
> **Author response**
>
> We thank you for your time and feedback regarding our work. We look forward to discussing several follow-up points provided below.
>
> > The main weakness is that no comparison of ZSClip-iqa with recognition-aware quality metrics is given.
>
> This is a fair point by the reviewer and our main point of clarification is that while we do introduce ZSCLIP-IQA as a new metric, we do so in the context of showing how the proposed causal framework (Section 3) provides a means to identify how new metrics can be developed that meet the criteria we described in Section 3.1.  Our main focus in the paper is not with developing a new metric but instead to provide a framework for identifying if/when IQ metrics can be informative of DNN performance.
>
> We wish to highlight a couple of additional points of clarification here and in the final manuscript:
>
> 1. We also want to emphasize that with this work, we are the first to establish a clear instance where traditional IQA metrics calibrated toward human perceptual judgements do not provide quality scores that good predictors of model performance.  We use the causal framework to provide theoretical and empirical arguments for why this might be the case in the context of image classification.
> 2. With ZSCLIP-IQA, we will clarify in the manuscript that we do not claim to be the first nor the best recognition-aware quality metric.
> 3. [minor] The works cited do provide a good perspective on aligning IQA with performance, however the quality metrics in those papers are mainly trained/tested on partitions of the same datasets which makes these methods closer to the strong task-guided methods we presented in Section 5.
>
> We do agree that if we wish to improve and demonstrate broad generality of ZSCLIP-IQA, a deeper comparison with methods such as those cited would be warranted in future work.
>
>
> > What is the relation of the proposed metric and the existing IQA metrics designed for face/image classification accuracy?
>
> The primary purpose of the proposed metric is really to show how the causal framework outlined in Section 3 can be used to design IQ metrics that satisfy the four desiderata (as demonstrated with the results in Table 4).  While existing IQA metrics are typically calibrated to human perceptual judgements, it was unclear whether they would be correlated with DNN performance (i.e., Fig. 1).  Using a causal model such as in Fig 6. to develop the ZSCLIP-IQA metric, we could show that, by construction, $Q$ is predictive of $M$ and satisfies all desiderata.
>
> As mentioned above, we will be clear in the manuscript that we're not claiming to be the first nor the best IQ metric aligned with task performance, but we do show how a metric developed using our causal framework can be a viable solution to the IQ-performance alignment problem. We will clarify this in the manuscript and make clear that we're not making broader claims about the applicability of ZSCLIP-IQA for other tasks/datasets.
>
>
> > What is the transferability of the proposed metric for predicting DNN performance in tasks other than classification?
>
> As discussed in the limitations portion of Section 7, while the causal framework and desiderata that we've presented are extensible to other tasks, we leave it to future work to explore this question.  In fact, our formulation relies on no asumptions about the task or domain and should generalize. In practice, we expect different tasks will influence the *degree* to which $Q$ can be predictive of $M$, but the causal framework at least ensures we can identify the conditions under which $Q$ satisfies the four desiderata (Sec 3.1) regardless of the task or domain.
>
> Also, we focused on classification initially because showing that the notion of quality is task-dependent, even in just the classification context, will naturally have a broad impact given the significant attention that image classification has received over the last decade in computer vision and machine learning research and applications.

---

> > ### Comment · Reviewer_xKhd · 2024-11-26
> > **Reviewer feedback to author response**
> >
> > Thank you for your comments; however, I am not convinced by the contribution. Given that you propose a framework for developing IQ metrics aligned with DNN performance, you should compare it with the existing solutions. Take a look at the existing problem formulation in IQA. Most problem formulations for IQA metrics require a high correlation with human scores, yielding a low correlation of these metrics with DNN performance. This is what you partially showed in the paper. However, some metrics are aligned with DNN performance. Can they be classified somehow within the proposed framework? I suggest that with the proposed framework, you need to either classify the existing IQ metrics (or their problem formulation) or show why the framework will be helpful for the community. For example, if ZSCLIP-IQA or its variations beat existing recognition-aware metrics, then it is clear that the framework helps to develop metrics of new types.

---

> ### Author Response · Authors · 2024-11-27
>
> Thank you for these clarifying points. We do agree that in order for the framework to be generalizable, it would be valuable to at least show how it can be used to classify other quality metrics.  We will add to the manuscript more details on how the works the reviewer cited could be situated relative to ZSCLIP-IQA and traditional NR-IQA metrics.
>
> While beating other task-aware metrics would be a valuable outcome, this was not the objective of this work. Our primary goal was to investigate whether traditional NR-IQA metrics correlated with DNN performance. When we found there was low correlation (which has not previously been investigated in depth), we developed the causal framework that enables **bridging the gap** between traditional and task-aware IQ metrics while presenting multiple approaches (i.e., strong/weak task-guided metrics) along that spectrum. In this context, we would still argue that showing a new state-of-the-art task-aware IQ metric is not necessary to demonstrate (1) that NR-IQA metrics are not informative of task DNN performance and (2) that the framework can be used to establish the link between IQ metrics and DNN performance (as shown through the strong and weak task-guided metrics).  As suggested, we will modify the manuscript to at least show how existing task-aware metrics could be formulated in our framework.

---

### Meta-Review · Area_Chair_7PUp · 2024-12-21

**Metareview:**

The paper introduces a causal framework to analyze the alignment of IQA metrics with DNN classifier accuracy, find that IQA metrics do not generally align with task performance, and proceed to propose a new IQA metric, ZSCLIP-IQA, which is better aligned.

While the definition of the framework itself is reasonable, the main issue with this paper is the lack of empirical validation. To be useful to the community, the framework must correctly explain the performance of existing models. In particular, reviewer xKhd points out existing recognition-aware quality metrics that should be accounted for in this paper. Reviewer qL7W comments that the paper is focused too much on object recognition and lacks evaluations on other image analysis tasks. Reviewer KDbi criticizes the lack of diversity in image distortions used to evaluate performance.

Overall, I have decided to reject this paper, as not enough evidence is provided to show that the framework is as general and useful as claimed.

**Additional Comments On Reviewer Discussion:**

While reviewers came to different conclusions in terms of their rating, all of them brought up issues with the empirical evaluation. The authors did not sufficiently address these issues, for example by adding more experimental data.

---

### Decision · Program_Chairs · 2025-01-22

Reject